# A Secured Proxy-Based Data Sharing Module in IoT Environments Using Blockchain

**DOI:** 10.3390/s19051235

**Published:** 2019-03-11

**Authors:** Kwame Opuni-Boachie Obour Agyekum, Qi Xia, Emmanuel Boateng Sifah, Jianbin Gao, Hu Xia, Xiaojiang Du, Moshen Guizani

**Affiliations:** 1Center for Cyber Security, University of Electronic Science and Technology of China, Chengdu 611731, China; obour539@yahoo.com (K.O.-B.O.A.); emmanuelsifah@yahoo.com (E.B.S.); xiahu@uestc.edu.cn (H.X.); 2CETC Big Data Research Institute Co., Ltd., Guiyang 550008, China; 3School of Resources and Environment, Center for Digital Health, University of Electronic Science and Technology of China, Chengdu 611731, China; gaojb@uestc.edu.cn; 4Department of Computer and Information Sciences, Temple University, Philadelphia, PA 19122, USA; dxj@ieee.org; 5Department of College of Engineering, Qatar University, Doha, Qatar; mguizani@ieee.org

**Keywords:** Attribute-Based Encryption (ABE), blockchain, cyber-security, fine-grained access control, Inner-Product Encryption (IPE), Internet of Things (IoT), proxy re-encryption

## Abstract

Access and utilization of data are central to the cloud computing paradigm. With the advent of the Internet of Things (IoT), the tendency of data sharing on the cloud has seen enormous growth. With data sharing comes numerous security and privacy issues. In the process of ensuring data confidentiality and fine-grained access control to data in the cloud, several studies have proposed Attribute-Based Encryption (ABE) schemes, with Key Policy-ABE (KP-ABE) being the prominent one. Recent works have however suggested that the confidentiality of data is violated through collusion attacks between a revoked user and the cloud server. We present a secured and efficient Proxy Re-Encryption (PRE) scheme that incorporates an Inner-Product Encryption (IPE) scheme in which decryption of data is possible if the inner product of the private key, associated with a set of attributes specified by the data owner, and the associated ciphertext is equal to zero $\left(0\right)$. We utilize a blockchain network whose processing node acts as the proxy server and performs re-encryption on the data. In ensuring data confidentiality and preventing collusion attacks, the data are divided into two, with one part stored on the blockchain network and the other part stored on the cloud. Our approach also achieves fine-grained access control.

## 1. Introduction

It has been estimated that there will be an enormous growth in the number of devices that will be connected to the internet by 2030 [1], and this will diminish the boundary between physical and digital worlds [2]. Human populace is not the main driver for this growth, but rather it is as a result of advances in wireless communication, embedded computing technologies, actuation and sensing that allow devices in a cyber physical world to become connected entities. The Internet of Things (IoT) is expected to fundamentally transform human daily activities, thereby outlining human-to-machine (H2M), machine-to-machine (M2M) and human-to-human (H2H) interactions in the connected world. Services provided by the IoT, which ensure safety, can be thought of as real drivers towards a better world of connectivity, as expressed by the authors of [3]. A complex task is the development of IoT systems and IoT services, which in particular is a crucial activity that requires an in-depth research effort. In that essence, Casadei et al. [3] presented an “Opportunistic IoT Service” that extends the already existing IoT service models and considers some essential features for service provisioning. The authors of [4] also developed a toolset that synthesizes and validates human motion data aggregated from wearable computing devices, with the aim of enhancing the privacy of data owners. The toolset is developed to alleviate some of the challenges in data collection from IoT devices, and algorithm development. Their platform offers all the capabilities of existing datasets as well as enables the synthesis of data streams for users, scenarios and activities of their preference. Their proposal is cost effective and provides more extensive data for validation and system refinement.

Although these IoT platforms offer numerous opportunities and provide effective solutions for cyber-security, there are several challenges associated with the IoT. Providing a secured data sharing environment and also ensuring privacy, as the massive volumes of data generated by the IoT devices (either single devices or entire systems) are very sensitive, are a few of these challenges. Data owners tend to worry about how their data are used since the control is out of their hands. The security of their data is the most prominent issue in cloud computing, and this affects the performance of this paradigm [5,6]. Encrypting data before outsourcing them has promised to be a good way in mitigating the security concerns [7]. When the data are encrypted, it becomes difficult to share the data with users because the owner has to share the decryption key with those users, thereby granting access to the data. Another problem that arises from the sharing of keys is user revocation, where there should be denial of service for some users. What data owners usually do is invalidate the existing key by performing a re-encryption over the whole set of data with a new key, and in turn re-distributing the key to the (authorized) users. This action also becomes cumbersome and enormously involving when there are huge amounts of data to outsource, and the owner does not keep a copy of the outsourced data locally. Due to the generation and management of keys and ciphertexts, the provision of access control on encrypted data also becomes a challenge.

Attribute-based encryption (ABE), an encryption scheme first proposed by Sahai and Waters [8], achieves both access control and data security by granting different access rights to users based on their attributes. One of its characteristics is the revocation of access rights to users. The use of attribute-based encryption also aims at providing fine-grained access control, as it determines which user has access rights to which kind of data. ABE is an ideal tool to realize complex access control policies: the data to be accessed is associated with a set of attributes, and the privileges of the user are specified by a logical expression over these attributes. Other studies [9,10] also propose methods to attain fine-grained access control. Some fine-grained access control modules, as proposed by Yu et al. [11], however, place a heavy burden on the cloud (although believed to be more powerful than the data owner). The cloud, as a single point of service, is always expected to serve a greater number of users and therefore it becomes imperative to have as minimal overhead as possible to the cloud. Cloud service providers may also charge data owners on the amount of computations they may have to make. Therefore, the lower the computation, the lesser the cost. To ensure effective data sharing and user revocation, a system model employing Key Policy-Attribute Based Encryption (KP-ABE) and Proxy Re-Encryption (PRE) is proposed in [11]. The proposed scheme is, however, vulnerable to collusion attacks using a revoked user and the cloud server.

In this work, we incorporate the decentralized and consensus-driven blockchain technology and its underlying cryptographic primitives, and proxy re-encryption mechanism based on ABE to actualize the confidentiality of data. In this system, the data after re-encryption, which is done by the blockchain’s processing node (a trusted proxy), are divided into two with one part stored on the blockchain and the other part on the cloud server. Therefore, it becomes impractical for a revoked user to obtain the data even after colluding with the honest, but curious cloud server. Furthermore, to lessen the burden on the cloud server, computations on the data are performed by the blockchain’s processing nodes. We therefore propose a proxy re-encryption scheme based on a well-established ABE scheme proposed by Park [12].

It’s true edge computing brings a better satisfaction to IoT devices. However, the storage of the data is done on the cloud and not much processing is done by/on the cloud. All processes are executed by the blockchain processing nodes, which have more computing power than the resource-constrained IoT devices. Moreover, the protocol still works, be it on the cloud or at the edge, since the main focus of this paper is on the security scheme. Due to constraints on resources by the IoT devices, the implementation of the security model, which is part of the computations and processing, is done by the blockchain network because it has enough processing power. Therefore, there are no specific hardware/software requirements for the resource-constrained IoT devices.

To summarize, our proxy re-encryption satisfies fine-grained access control in that users have access right to different sets of data, which is made possible by the ABE scheme. Our scheme is also collusion resistant as the cloud server and/or the proxy and the (revoked) user cannot collude to access data. This is made possible because the blockchain network is a decentralized system and all processes (transactions) are monitored by every participant on the network, and also recorded and stored into blocks. Furthermore, there is an appreciable level of trust between the data owner and the users due to the utilization of blockchain, as it ensures a trustworthy environment among participants involved. The proxy is uni-directional, as it transforms a ciphertext *C* into a ciphertext $C^{\prime }$ in only one direction, but not in the reverse transform.

The remainder of this paper is organized as follows. In Section 2, related works on the cryptographic primitives, IoT and blockchain are reviewed. In Section 3, we introduce the notations to be used in this paper, while the system model is formulated in Section 4. Our proposed scheme and its security model are presented in Section 5 and Section 6, respectively. Implementation and performance analysis are presented in Section 7, while Section 8 provides a set of discussions. Section 9 concludes the paper.

## 2. Related Works

The secured sharing of data among several users via a cloud service provider is extensively researched in [13,14,15]. Mambo and Okamoto’s [16] novel PRE scheme has been adopted as the technique to achieve this, and it was further extended by Blaze et al. [17] by basing their findings on the El-Gamal cryptosystem [18]. In their work, a proxy can transform a message encrypted under Alice’s key into an encryption of the same message under Bob’s key because it utilizes a re-encryption key. While effective data sharing can be achieved by these schemes by meeting some security requirements and properties, there is no enforcement of fine-grained access control on the shared data.

Attribute-based proxy encryption techniques [19,20,21,22] have therefore been adopted to enforce this. Both the ciphertext and the private key of the user are associated with an attribute set in the ABE scheme, and decryption is possible when there is a match between the set of attributes for both the private key and the ciphertext [8,23,24]. These approaches, nevertheless, help only in the adversary not obtaining any information about the encrypted message. Katz et al. [25] therefore presented an attribute hiding scheme for a class of predicates. This was known as Inner-Product Encryption (IPE), and it preserves the confidentiality of the attributes associated with the ciphertext. Following that, a hierarchical IPE scheme that uses an *n*-dimensional vector space in bilinear maps of prime order was proposed by Okamoto et al. [26], and the full security under the standard model was achieved. Park [12] therefore presented an IPE scheme that supports an attribute hiding property, and also is secure against Decisional Bilinear Diffie–Hellman (D-BDH) and decisional linear assumptions.

Du et al. [27] presented an efficient and scalable key management scheme for heterogeneous sensor networks. Their scheme utilizes the fact that there is a lower communication and computational cost when a sensor only communicates with a small portion of its neighbors. An Elliptic Curve Cryptographic (ECC) scheme is used to further improve key management, as it also reduces sensor storage requirement and energy consumption while achieving better security. Xiao et al. [28] surveyed the various techniques utilized in the key management for Wireless Sensor Networks (WSNs). Their survey paper looks at both the advantages and disadvantages of the various techniques. It is realized that no key distribution technique is ideal to all the scenarios where the sensor networks are deployed, and therefore the technique being employed should meet the requirements of both the application in question and the resources of the individual sensor networks. The authors of [29] presented an effective key management scheme for heterogeneous sensor networks, which is quite similar to the work in [27]. Their work portrays how efficient the performance of their scheme is, and that it significantly achieves a better security than existing sensor network key management schemes. Du et al. in [30] presented the security issues in WSNs. Quite similar to the aforementioned sensor-related papers, they investigated schemes that achieve better security and also lower computational cost for the sensor networks.

Blockchain technology offers a suitable platform that can be used for numerous applications in medical care. Improving the security in medical data sharing and automating the delivery of health-related notifications are the massive potentials of this technology, and they are compatible with the Health Insurance Portability and Accountability Act (HIPAA) [31]. Several authors have provided blockchain health-related applications [32,33,34,35]. The authors of [32] determined the current challenges of Electronic Medical Record (EMR) systems and the potential they have in providing solutions to security challenges and interoperability, with the use of blockchain technology. Focus has been on the application of blockchain to Electronic Health Records (EHRs) to facilitate interoperability. Medrec, a prototype released by MIT, expresses a practical way of sharing healthcare data between EHRs and blockchain [33]. A secure and scalable access control system for confidential information sharing on blockchain was also presented by the authors of [34]. Their results portray the effectiveness of their system in instances where traditional methods of access control failed. Yue et al. designed a concept for an application that presents patients with the opportunity to grant access to information about their health records to designated individuals [35]. The authors of [36] proposed a novel protocol that achieves patient privacy preservation by applying the concept of blockchain in an eHealth platform.

A possible efficient data sharing platform among interested parties and the preservation of privacy are a just few of the opportunities blockchain technology offers. For blockchain to reach its maximum potential, it is essential to tackle one of the most important problems facing this technology: data access control. This work therefore places more emphasis on providing a secured data access control in a data sharing environment. A blockchain processing node acts as a proxy and performs re-encryption on data that are given to a secondary user. Our system preserves data confidentiality and integrity, and avoids collusion attacks. Fine-grained access control is also achieved.

## 3. Preliminaries

We introduce some of the notations that will be utilized throughout this paper in this section.

### 3.1. Bilinear Maps

Our protocol is based on bilinear maps [37]. Let *G* and $G_{T}$ be two multiplicative cyclic groups that have a prime order *p*, and *g* be a generator of *G*. A bilinear map $e:G\times G\to G_{T}$ has the following properties:Bilinear: For all $a,b\in Z_{p}$, g,h∈G, then $e\left(g^{a},h^{b}\right)=e{\left(g,h\right)}^{ab}$ can be computed efficiently.The map is non-degenerate. That is, if *g* generates *G* and *h* also generates *G*, then e(g,h) generates $G_{T}$. In addition, e(g,h)≠1. The map does not send all pairs in G×G to the identity in $G_{T}$.It is computable; there exists an efficient algorithm to compute the map e(g,h) for any g,h∈G.

Note that e(,) is symmetric since $e\left(g^{a},h^{b}\right)=e{\left(g,h\right)}^{ab}=e\left(g^{b},h^{a}\right)$.

### 3.2. Inner-Product Encryption (IPE)

The Inner Product Encryption (IPE) scheme, as proposed in [12], is an attribute-based encryption technique in which both ciphertext(s) and private (secret) key(s) are associated with vectors. Access to and decryption of an encrypted data can only be possible if and only if the inner product of the private key, which is related to vector $\vec{v}$, and the ciphertext, also related to vector $\vec{x}$, is 0. That is, for the two vectors, $\left(\vec{v}\cdot \vec{x}\right)=\sum_{i=1}^{n}x_{i}\cdot v_{i}\;\mathrm{mod}\;p=0$. Let ∑ be a set of attributes peculiar to particular encrypted data that involves vector $\vec{v}$ and has a dimension of *n*. Denote *F* as representing a predicate class that involves an inner product over vectors, i.e., $F=f_{\vec{x}}|\vec{x}\in \sum$ such that $f_{\vec{x}}\left(\vec{v}\right)=1$ iff $(\vec{x}\cdot \vec{v})=0$. Two *n*-dimensional vectors, $\vec{x}=\left(x_{1},\ldots ,x_{n}\right)$ and $\vec{v}=\left(v_{1},\ldots ,v_{n}\right)$, all belonging to the set of attributes, ∑, are, respectively, utilized in the encryption and key decryption phases.

We incorporate the rationale behind a proxy’s re-encryption key (RE key) into this work by using the IPE scheme to transform a ciphertext associated with a vector into a new ciphertext associated with another vector but encrypts the same message (m∈M). We ensure that there is no revealing of the information about the encrypted data.

### 3.3. Attribute Based Encryption (ABE)

There are two main classifications of ABE schemes, namely Ciphertext Policy-Attribute Based Encryption (CP-ABE) [23] and Key Policy-Attribute Based Encryption (KP-ABE) [38]. In this paper, we make use of KP-ABE, as the data are encrypted by a set of attributes and the private keys of the users are associated with the access structure of KP-ABE. Thus, if the attribute of the encrypted data satisfies the access structure of the user’s private key, decryption of the ciphertext can occur.

### 3.4. Proxy Re-Encryption (PRE)

The notion of “atomic proxy cryptography” is the basis for proxy re-encryption, which was first introduced by Mambo and Okamoto [16]. This scheme basically makes use of a semi-trusted proxy that transforms the ciphertext for Alice into a ciphertext for Bob, without actually knowing or gaining access to the plaintext. Popular, well-known proxy re-encryption schemes are the Blaze, Bleumer and Strauss (BBS) [17] and the Ateniese, Fu, Green and Hohenberger (AFGH) [39] schemes, which are based on El Gamal and Bilinear maps cryptographic algorithms, respectively. In this work, the blockchain processing node (a trusted entity) serves as the proxy, and performs re-encryption on the data.

### 3.5. Blockchain Network

Blockchain technology, originally proposed by Satoshi Nakamoto [40], acts as a shared, decentralized ledger to record transactions. Public, private and consortium blockchains are the three main types of blockchain. For decentralized networks and offering transparency, public blockchain is predominantly used. Private and consortium blockchains are, however, preferred when more control and privacy are of the essence. Consensus and decentralization, key features of blockchain, are the reasons for using blockchain technology in our system. Moreover, our blockchain’s processing node serves as the trusted proxy that performs the re-encryption on the data before they are given to the secondary user. Proof-of-work (PoW) and Practical Byzantine Fault Tolerance (PBFT) provide security offered by the use of this technology. They utilize the agreement of nodes in the addition of a block to the chain, which acts as a ledger for all transactions.

Blockchain has helped in the effectiveness and advancement of many industries. It is also capable of implementing smart contracts, which are programmable scripts that automatically execute actions based on pre-defined triggers. The smart contracts are called upon when a data user requests access to data. Prior to the data being sent to the cloud, the owner specifies how its data are to be used and gives the details to the blockchain network. A processing node then embeds the contract into the data being given to the requestor. Our blockchain keeps logs of the transactions to achieve effective auditing.

Due to privacy concerns, our system utilizes the distributed ledger property of the blockchain, namely immutability, for authenticity and verifiability, and also the use of the consortium blockchain. Only authorized users can gain access to data. This enhances transparency for data owners, and allows them to effectively manage their data.

A block consists of a single event, with the event spanning from the time a request is made to when the block is broadcast onto the blockchain. Consensus nodes are responsible for mining and reporting all activities. A block is made up of a format that distinctively describes the block. This is followed by a block size, and then a block header, which is hashed with sha256(sha256()) as implemented in Bitcoin headers [40]. The block size contains the size of the block and the header ensures immutability. Changing a block header, in order to falsify a piece of information, requires a change to all headers starting from the genesis (parent) block.

A block header also contains the version number which indicates the validation rules to follow. The previous block’s hash is also contained in the header. A timestamp is also included in the header and it indicates when the block was created. A target difficulty, which is a value that indicates how processing is achieved by the consensus nodes, is also found in the header. This makes processing difficult for malicious nodes but solvable by verified consensus nodes. There is also an arbitrary number generated by the consensus nodes, which modifies the header hash in order to produce a hash below the target difficulty. This is called a nonce. A transaction counter is found in the block, whose function is to record the total number of transactions in the entire block. The transaction is made up of the consensus transaction and the user transaction. Each type comprises a timestamp and the data. A block locktime defines the structure for the entire block. This is a timestamp that records the last entry of a transaction as well as the closure of a block. When all conditions are met, the block is then broadcast onto the blockchain network.

For scalability concerns, our blockchain stores hashes of transactions. Transactions on this blockchain typically include data requests, data processing (encryption and/or re-encryption), and data access.

## 4. System Model

### 4.1. Problem Statement

We demonstrate a simple IoT file/data sharing scenario in a healthcare environment for the sake of clarity, where we consider a patient whose data can only be accessed by his/her physician, pharmacist or relatives. Patients’ data are normally collected and collated by health sensors that are usually bound to them, and uploaded onto a cloud server after recording. Before a patient’s medical data are outsourced to a cloud server, the patient encrypts their own data under a set of attributes, which indicates the access privilege on the data. The patient then gives the details of all authorized users to the blockchain’s processing node. Thus, access to a patient’s data can be possible only if the user satisfies the attribute set and also uses the private key related to that attribute set.

However, there may be an instance where a physician might share the patient’s data, depending on the kind of ailment they are treating, with other healthcare professionals who are not in the same hospital and therefore have a different access policy on the data. It now lies of the proxy (blockchain processing node) to re-encrypt the patient’s data under the patient’s attribute set to the new attribute set in a way that does not reveal any information about the data and its corresponding attributes. This must also be done in an efficient and secured way. The model of our system is presented in Figure 1.

### 4.2. System Overview

**Data Owner**: This is the entity (the patient in this case) whose data are to be accessed. Access is possible if and only if the private key of the data user corresponds to the attribute set specified by the data owner.**Data User**: This is the entity who wants to make use of the data from the owner. Both the data owner and user(s) should be registered on the blockchain.**Cloud Server**: This is the repository for the data from the owner. All encrypted files are sent to the cloud server (honest, but curious) through a secured communication channel.**Blockchain Network**: This primarily consists of the following entities:
Issuer: This entity registers the participants (data owner and users) on the blockchain network. It gives out membership keys to them and that serves as their identity (ID).Verifier: The verifier, which also serves as an authentication unit, checks whether a user who makes an access request or a data owner who uploads its data onto the cloud, are actually members of the blockchain network.Processing node: This is the heartbeat of the blockchain network. All processes (transactions) that ever occur on the network are performed by this entity. In this work, however, it serves as the (trusted) proxy that oversees the re-encryption process.Smart contract center: This unit prepares the contract that binds how data are to be used.

The various processes that happen in the system model are described below:The proxy generates a secret key, SK, and a public key $P_{pub}$, and hands the public key and access policy to the data owner. That is, the data owner is given $\{P_{pub},H_{access}\}$.The patient encrypts the data with the attribute set and sends the encrypted data to the cloud through a secured channel. The encrypted data are $CT=\{Enc(M,\vec{x})\}$.The data user makes a request for the data.The proxy accesses the permission rights of the data users from the cloud server. After accessing it, the blockchain network, which also serves as a trusted authority, gives the private key to the user according to the user’s attributes.Users can now access data from the cloud server.The primary user is given $PK_{\vec{v}}$ while the secondary user is given $PK_{\vec{v^{\prime }}}$. The proxy generates a re-encryption key REKey and transforms the policy set $H\to H^{\prime }$ for the secondary user who wants the shared data from the primary user but holds a different access policy, $H^{\prime }$.

## 5. The Scheme

As in several security algorithms, our proposed scheme consists of the following algorithms: *Setup*, *KGen*, *Encrypt*, *RKGen*, *ReEncrypt*, and *Decrypt*. The IPE scheme, as presented in [12], is adopted in this work and therefore most of the algorithms will be the same. *Setup*, *KGen*, *Enrypt* and *Decrypt* have been previously presented in [12].

The assumption is made here that $\sum ={\left(Z_{p}\right)}^{n}$ is the set of attributes bound to data, where *n* is the dimension of the vectors, $\vec{x}$ and $\vec{v}$, and *p* is the prime order of the group, *Z*. For any vector $\vec{v}=\left(v_{1},\ldots ,v_{n}\right)\in \sum$, each element $v_{i}$ belongs to the set $Z_{p}$. The algorithms are as follows.

$\left(P_{pub},SK\right)$ ← **Setup**
(λ,n): With any security parameter $\lambda \in Z^{+}$, the setup algorithm runs σ(λ) after which a tuple $\left(p,G,G_{2},e\right)$ is obtained. A random generator g∈G, along with random exponents $\delta_{1}$, $\delta_{2}$, $\theta_{1}$, $\theta_{2}$, ${\left\{w_{1,i}\right\}}_{i=1}^{n}$, ${\left\{t_{1,i}\right\}}_{i=1}^{n}$, ${\left\{f_{1,i},f_{2,i}\right\}}_{i=1}^{n}$ and ${\left\{h_{1,i},h_{2,i}\right\}}_{i=1}^{n}$, found in $Z_{p}$ are all selected. A random element, $g_{2}\in G$, is also selected. Furthermore, it selects a random number, $\Psi \in Z_{p}$ and obtains the set of elements ${\left\{w_{2,i}\right\}}_{i=1}^{n}$, ${\left\{t_{2,i}\right\}}_{i=1}^{n}$ in $Z_{p}$ with constraints such that
$$\Psi =\delta_{1}w_{2,i}-\delta_{2}w_{1,i},\Psi =\theta_{1}t_{2,i}-\theta_{2}t_{1,i}$$

The setup algorithm then computes
$$W_{1,i}=g^{w_{1,i}},W_{2,i}=g^{w_{2,i}},T_{1,i}=g^{t_{1,i}},T_{2,i}=g^{t_{2,i}},F_{1,i}=g^{f_{1,i}},F_{2,i}=g^{f_{2,i}},H_{1,i}=g^{h_{1,i}},H_{2,i}=g^{h_{2,i}}$$

Now, the following notations are also given:$$Q_{1}=g^{\delta_{1}},Q_{2}=g^{\delta_{2}},R_{1}=g^{\theta_{1}},R_{2}=g^{\theta_{2}},g_{1}=g^{\Psi },\Upsilon =e\left(g,g_{2}\right)$$

The public $P_{pub}$ and secret SK keys are then, respectively, computed as:$$P_{pub}=\left(g,g^{\Psi },{\left\{W_{1,i},W_{2,i},F_{1,i},F_{2,i}\right\}}_{i=1}^{n},{\left\{T_{1,i},T_{2,i},H_{1,i},H_{2,i}\right\}}_{i=1}^{n},{\left\{Q_{i},R_{i}\right\}}_{i=1}^{2},\Upsilon \right)\in G^{8n+6}\times G_{T}$$
$$SK=\left({\left\{w_{1,i},w_{2,i},t_{1,i},t_{2,i},f_{1,i},f_{2,i},h_{1,i},h_{2,i}\right\}}_{i=1}^{n},{\left\{\delta_{i},\theta_{i}\right\}}_{i=1}^{2},g_{2}\right)\in Z_{p}^{8n+4}\times G$$

$PK_{\vec{v}}$ ← **KGen**
$\left(SK,\vec{v}\right)$: For a vector $\vec{v}=\left(v_{1},\ldots ,v_{n}\right)$, the algorithm selects random exponents $\lambda_{1}$, $\lambda_{2}$, ${\left\{r_{i},\phi_{i}\right\}}_{i=1}^{n}$ in $Z_{p}$, and creates a private key $PK_{\vec{v}}=\left(K_{A},K_{B},{\left\{K_{1,i},K_{2,i}\right\}}_{i=1}^{n},{\left\{K_{3,i},K_{4,i}\right\}}_{i=1}^{n}\right)$
$\in G^{4n+2}$. The composition of the various elements in the $PK_{\vec{v}}$ is defined as follows: $${\left\{K_{1,i}=g^{-\delta_{2}r_{i}}g^{\lambda_{1}v_{i}w_{2,i}},K_{2,i}=g^{\delta_{1}r_{i}}g^{-\lambda_{1}v_{i}w_{1,i}}\right\}}_{i=1}^{n},{\left\{K_{3,i}=g^{-\theta_{2}\phi_{i}}g^{\lambda_{2}v_{i}t_{2,i}},K_{4,i}=g^{\theta_{1}\phi_{i}}g^{-\lambda_{2}v_{i}t_{1,i}}\right\}}_{i=1}^{n},$$
$$K_{A}=g_{2}\prod_{i=1}^{n}K_{1,i}^{-f_{1,i}}K_{2,i}^{-f_{2,i}}K_{3,i}^{-h_{1,i}}K_{4,i}^{-h_{2,i}},K_{B}=\prod_{i=1}^{n}g^{-(r_{i}+\phi_{i})}$$

CT ⟵ **Encrypt**
$\left(P_{pub},\vec{x},M\right)$: To encrypt a message $M\in G_{T}$ and a vector $\vec{x}=\left(x_{1},\ldots ,x_{n}\right)\in \left(Z_{p}\right)$ under the public key $P_{pub}$, the algorithm selects random elements ${\left\{s_{i}\right\}}_{i=1}^{4}\in Z_{p}$ and uses them to compute the ciphertext CT as follows:$$CT=(g^{s_{2}},g_{1}^{s_{1}},{\left\{W_{1,i}^{s_{1}}\cdot F_{1,i}^{s_{2}}\cdot Q_{1}^{x_{i}s_{3}},W_{2,i}^{s_{1}}\cdot F_{2,i}^{s_{2}}\cdot Q_{2}^{x_{i}s_{3}}\right\}}_{i=1}^{n},{\left\{T_{1,i}^{s_{1}}\cdot H_{1,i}^{s_{2}}\cdot R_{1}^{x_{i}s_{4}},T_{2,i}^{s_{1}}\cdot H_{2,i}^{s_{2}}\cdot R_{2}^{x_{i}s_{4}}\right\}}_{i=1}^{n},$$
$${\left\{T_{1,i}^{s_{1}}\cdot H_{1,i}^{s_{2}}\cdot R_{1}^{x_{i}s_{4}},T_{2,i}^{s_{1}}\cdot H_{2,i}^{s_{2}}\cdot R_{2}^{x_{i}s_{4}}\right\}}_{i=1}^{n},\Upsilon^{-s_{2}}M)\in G^{4n+2}\times G_{T}$$
where $s_{2}=\Psi s_{1}$.

$REKey_{\vec{v}}$ ← **RKGen**
$\left(SK,\vec{v},\vec{x^{\prime }}\right)$: **KGen** algorithm is first called and a random element, $l\in Z_{p}$, is selected. It then computes α, $\alpha^{\delta_{2}}$, $\alpha^{-\delta_{1}}$, $\alpha^{\theta_{2}}$, and $\alpha^{-\theta_{1}}$, where $\alpha =g_{2}^{l}$. The **Encrypt** algorithm is then called to encrypt α under the vector $\vec{x^{\prime }}$ by utilizing $Encrypt(P_{pub},\vec{x^{\prime }},\alpha )$. The output is a ciphertext $CT_{A}$. The RKGen algorithm then selects random exponents ${\left\{\lambda_{i}^{\prime }\right\}}_{i=1}^{2}$, ${\left\{r_{i}^{\prime },\phi_{i}^{\prime }\right\}}_{i=1}^{n}$
$\in Z_{p}$ and uses them to compute $REKey_{\vec{v}}$ as follows: $${\left\{K_{1,i}^{\prime }=g^{-\delta_{2}r_{i}^{\prime }}g^{\lambda_{1}^{\prime }v_{i}w_{2,i}}\alpha^{\delta_{2}},K_{2,i}^{\prime }=g^{\delta_{1}r_{i}^{\prime }}g^{-\lambda_{1}^{\prime }v_{i}w_{1,i}}\alpha^{-\delta_{1}}\right\}}_{i=1}^{n},{\left\{K_{3,i}^{\prime }=g^{-\theta_{2}\phi_{i}^{\prime }}g^{\lambda_{2}^{\prime }v_{i}t_{2,i}}\alpha^{\theta_{2}},K_{4,i}^{\prime }=g^{\theta_{1}\phi_{i}^{\prime }}g^{-\lambda_{2}^{\prime }v_{i}t_{1,i}}\alpha^{-\theta_{1}}\right\}}_{i=1}^{n},$$
$$K_{A}^{\prime }=g_{2}\prod_{i=1}^{n}K_{1,i}^{{}^{\prime }-f_{1,i}}K_{2,i}^{{}^{\prime }-f_{2,i}}K_{3,i}^{{}^{\prime }-h_{1,i}}K_{4,i}^{{}^{\prime }-h_{2,i}},K_{B}^{\prime }=\prod_{i=1}^{n}g^{-(r_{i}^{\prime }+\phi_{i}^{\prime })}$$

$CT^{\prime }$ ⟵ **ReEncrypt**
$\left(REKey_{\vec{v}},CT\right)$: On input of the ciphertext CT and the re-encryption key $REKey_{\vec{v}}$, this algorithm first checks whether the attributes list of the user in $REKey_{\vec{v}}$ satisfies the attribute set of the CT. If that is not the case, it returns ⊥; else, ∀i={1,…,n}, the algorithm first computes the following:$$\prod_{i=1}^{n}e(C_{1,i},K_{1,i}^{\prime })\cdot e(C_{2,i},K_{2,i}^{\prime })\cdot e(C_{3,i},K_{3,i}^{\prime })\cdot e(C_{4,i},K_{4,i}^{\prime })$$

$$=\prod_{i=1}^{n}e\left(g^{w_{1,i}s_{1}}g^{f_{1,i}s_{2}}g^{\delta_{1}x_{i}s_{3}},g^{-\delta_{2}r_{i}^{\prime }}g^{\lambda_{1}^{\prime }v_{i}w_{2,i}}\alpha^{\delta_{2}}\right)\cdot e\left(g^{w_{2,i}s_{1}}g^{f_{2,i}s_{2}}g^{\delta_{2}x_{i}s_{3}},g^{\delta_{1}r_{i}^{\prime }}g^{-\lambda_{1}^{\prime }v_{i}w_{1,i}}\alpha^{-\delta_{1}}\right)$$

$$\cdot e\left(g^{t_{1,i}s_{1}}g^{h_{1,i}s_{2}}g^{\theta_{1}x_{i}s_{4}},g^{-\theta_{2}\phi_{i}^{\prime }}g^{\lambda_{2}^{\prime }v_{i}t_{2,i}}\alpha^{\theta_{2}}\right)\cdot e\left(g^{t_{2,i}s_{1}}g^{h_{2,i}s_{2}}g^{\theta_{2}x_{i}s_{4}},g^{\theta_{1}\phi_{i}^{\prime }}g^{-\lambda_{2}^{\prime }v_{i}t_{1,i}}\alpha^{-\theta_{1}}\right)$$

$$=\prod_{i=1}^{n}e\left(g^{w_{1,i}s_{1}},g^{-\delta_{2}r_{i}^{\prime }}\right)\cdot e\left(g^{f_{1,i}s_{2}},g^{-\delta_{2}r_{i}^{\prime }}g^{\lambda_{1}^{\prime }v_{i}w_{2,i}}\alpha^{\delta_{2}}\right)\cdot e\left(g^{\delta_{1}x_{i}s_{3}},g^{\lambda_{1}^{\prime }v_{i}w_{2,i}}\right)\cdot e\left(g^{w_{1,i}s_{1}},\alpha^{s_{2}}\right)\cdot e\left(g^{w_{2,i}s_{1}},g^{\delta_{1}r_{i}^{\prime }}\right)$$

$$\cdot e\left(g^{f_{2,i}s_{2}},g^{\delta_{1}r_{i}^{\prime }}g^{-\lambda_{1}^{\prime }v_{i}w_{1,i}}\alpha^{-\delta_{1}}\right)\cdot e\left(g^{\delta_{2}x_{i}s_{3}},g^{-\lambda_{1}^{\prime }v_{i}w_{1,i}}\right)\cdot e\left(g^{w_{2,i}s_{1}},\alpha^{-\delta_{1}}\right)\cdot e\left(g^{t_{1,i}s_{1}},g^{-\theta_{2}\phi_{i}^{\prime }}\right)$$

$$\cdot e\left(g^{h_{1,i}s_{2}},g^{-\theta_{2}\phi_{i}^{\prime }}g^{\lambda_{2}^{\prime }v_{i}t_{2,i}}\alpha^{\theta_{2}}\right)\cdot e\left(g^{\theta_{1}x_{i}s_{4}},g^{\lambda_{2}^{\prime }v_{i}t_{2,i}}\right)\cdot e\left(g^{t_{1,i}s_{1}},\alpha^{\theta_{2}}\right)\cdot e\left(g^{t_{2,i}s_{1}},g^{\theta_{1}\phi_{i}^{\prime }}\right)$$

$$\cdot e\left(g^{h_{2,i}s_{2}},g^{\theta_{1}\phi_{i}^{\prime }}g^{-\lambda_{2}^{\prime }v_{i}t_{1,i}}\alpha^{-\theta_{1}}\right)\cdot e\left(g^{\theta_{2}x_{i}s_{4}},g^{-\lambda_{2}^{\prime }v_{i}t_{1,i}}\right)\cdot e\left(g^{t_{2,i}s_{1}},\alpha^{-\theta_{1}}\right)$$

$$=\prod_{i=1}^{n}e\left(g^{-\delta_{2}w_{1,i}},g^{r_{i}^{\prime }s_{1}}\right)\cdot e\left(g^{s_{2}},{\left(g^{-\delta_{2}r_{i}^{\prime }}g^{\lambda_{1}^{\prime }v_{i}w_{2,i}}\alpha^{\delta_{2}}\right)}^{f_{1,i}}\right)\cdot e{\left(g,g\right)}^{\lambda_{1}^{\prime }\delta_{1}w_{2,i}x_{i}v_{i}s_{3}}\cdot e\left(g^{w_{1,i}s_{1}},\alpha^{\delta_{2}}\right)\cdot e\left(g^{\delta_{1}w_{2,i}},g^{r_{i}^{\prime }s_{1}}\right)$$

$$\cdot e\left(g^{s_{2}},{\left(g^{\delta_{1}r_{i}^{\prime }}g^{-\lambda_{1}^{\prime }v_{i}w_{1,i}}\alpha^{-\delta_{1}}\right)}^{f_{2,i}}\right)\cdot e{\left(g,g\right)}^{-\lambda_{1}^{\prime }\delta_{2}w_{1,i}x_{i}v_{i}s_{3}}\cdot e\left(g^{w_{2,i}s_{1}},\alpha^{-\delta_{1}}\right)\cdot e\left(g^{-\theta_{2}t_{1,i}},g^{\phi_{i}^{\prime }s_{1}}\right)$$

$$\cdot e\left(g^{s_{2}},{\left(g^{-\theta_{2}\phi_{i}^{\prime }}g^{\lambda_{2}^{\prime }v_{i}t_{2,i}}\alpha^{\theta_{2}}\right)}^{h_{1,i}}\right)\cdot e{\left(g,g\right)}^{\lambda_{2}^{\prime }\theta_{1}t_{2,i}x_{i}v_{i}s_{4}}\cdot e\left(g^{t_{1,i}s_{1}},\alpha^{\theta_{2}}\right)\cdot e\left(g^{\theta_{1}t_{2,i}},g^{\phi_{i}^{\prime }s_{1}}\right)$$

$$\cdot e\left(g^{s_{2}},{\left(g^{\theta_{1}\phi_{i}^{\prime }}g^{-\lambda_{2}^{\prime }v_{i}t_{1,i}}\alpha^{-\theta_{1}}\right)}^{h_{2,i}}\right)\cdot e{\left(g,g\right)}^{-\lambda_{2}^{\prime }\theta_{2}t_{1,i}x_{i}v_{i}s_{4}}\cdot e\left(g^{t_{2,i}s_{1}},\alpha^{-\theta_{1}}\right)$$

$$=\prod_{i=1}^{n}e\left(g^{\delta_{1}w_{2,i}-\delta_{2}w_{1,i}},g^{r_{i}^{\prime }s_{1}}\right)\cdot e\left(g^{\theta_{1}t_{2,i}-\theta_{2}t_{1,i}},g^{\phi_{i}^{\prime }s_{1}}\right)\cdot e\left(g^{s_{2}},K_{1,i}^{{}^{\prime }f_{1,i}}K_{2,i}^{{}^{\prime }f_{2,i}}K_{3,i}^{{}^{\prime }h_{1,i}}K_{4,i}^{{}^{\prime }h_{2,i}}\right)\cdot e\left(g^{-\delta_{1}w_{2,i}+\delta_{2}w_{1,i}},\alpha^{s_{1}}\right)$$

$$\cdot e{\left(g,g\right)}^{\left[\lambda_{1}^{\prime }\left(\delta_{1}w_{2,i}-\delta_{2}w_{1,i}\right)s_{3}+\lambda_{2}^{\prime }\left(\theta_{1}t_{2,i}-\theta_{2}t_{1,i}\right)s_{4}\right]x_{i}v_{i}}\cdot e\left(g^{-\theta_{1}t_{2,i}-\theta_{2}t_{1,i}},\alpha^{s_{1}}\right)$$

$$=\prod_{i=1}^{n}e\left(g^{\Psi },g^{r_{i}^{\prime }s_{1}}\right)\cdot e\left(g^{\Psi },g^{\phi_{i}^{\prime }s_{1}}\right)\cdot e\left(g^{-\Psi },\alpha^{s_{1}}\right)\cdot e{\left(g,g\right)}^{\Psi \left[\lambda_{1}^{\prime }s_{3}+\lambda_{2}^{\prime }s_{4}\right]\vec{x}\cdot \vec{v}}$$

$$\cdot e\left(g^{s_{2}},K_{1,i}^{{}^{\prime }f_{1,i}}K_{2,i}^{{}^{\prime }f_{2,i}}K_{3,i}^{{}^{\prime }h_{1,i}}K_{4,i}^{{}^{\prime }h_{2,i}}\right)\cdot e\left(g^{-\Psi },\alpha^{s_{1}}\right)$$

$$=e\left(g^{\Psi s_{1}},\prod_{i=1}^{n}g^{\left(r_{i}^{\prime }+\phi_{i}^{\prime }\right)}\right)\cdot e{\left(g,g\right)}^{\Psi \left[\lambda_{1}^{\prime }s_{3}+\lambda_{2}^{\prime }s_{4}\right]\vec{x}\cdot \vec{v}}\cdot e\left(g^{-\Psi },\alpha^{s_{1}}\right)\cdot e\left(g^{s_{2}},\prod_{i=1}^{n}K_{1,i}^{{}^{\prime }f_{1,i}}K_{2,i}^{{}^{\prime }f_{2,i}}K_{3,i}^{{}^{\prime }h_{1,i}}K_{4,i}^{{}^{\prime }h_{2,i}}\right)$$

After completing this computation, the algorithm then computes $CT_{B}$ as: $$CT_{B}=e\left(A,K_{A}^{\prime }\right)\cdot e\left(B,K_{B}^{\prime }\right)\cdot \prod_{i=1}^{n}e\left(C_{1,i},K_{1,i}^{\prime }\right)\cdot e\left(C_{2,i},K_{2,i}^{\prime }\right)\cdot e\left(C_{3,i},K_{3,i}^{\prime }\right)\cdot e\left(C_{4,i},K_{4,i}^{\prime }\right)$$
$$=e\left(g^{s_{2}},g_{2}\prod_{i=1}^{n}K_{1,i}^{{}^{\prime }-f_{1,i}}K_{2,i}^{{}^{\prime }-f_{2,i}}K_{3,i}^{{}^{\prime }-h_{1,i}}K_{4,i}^{{}^{\prime }-h_{2,i}}\right)\cdot e\left(g^{\Psi s_{1}},\prod_{i=1}^{n}g^{-\left(r_{i}^{\prime }+\phi_{i}^{\prime }\right)}\right)\cdot e\left(g^{\Psi s_{1}},\prod_{i=1}^{n}g^{\left(r_{i}^{\prime }+\phi_{i}^{\prime }\right)}\right)\cdot e\left(g^{-\Psi },\alpha^{s_{1}}\right)$$
$$\cdot e\left(g^{s_{2}},\prod_{i=1}^{n}K_{1,i}^{{}^{\prime }f_{1,i}}K_{2,i}^{{}^{\prime }f_{2,i}}K_{3,i}^{{}^{\prime }h_{1,i}}K_{4,i}^{{}^{\prime }h_{2,i}}\right)\cdot e{\left(g,g\right)}^{\Psi \left[\lambda_{1}^{\prime }s_{3}+\lambda_{2}^{\prime }s_{4}\right]\vec{x}\cdot \vec{v}}$$
$$=e\left(g^{s_{2}},g_{2}\right)\cdot e{\left(g,g\right)}^{\Psi \left[\lambda_{1}^{\prime }s_{3}+\lambda_{2}^{\prime }s_{4}\right]\vec{x}\cdot \vec{v}}\cdot e\left(g^{-\Psi },\alpha^{s_{1}}\right)$$
recalling that $A=g^{s_{2}}$, $B=g^{\Psi s_{1}}$, with $s_{2}=\Psi s_{1}$.

The re-encrypted ciphertext $CT^{\prime }$ therefore becomes the tuple $\left(A,B,CT_{A},CT_{B},D=e{\left(g,g_{2}\right)}^{-s_{2}}M\right).$

*M* ← **Decrypt**
$\left(CT,PK_{\vec{v}}\right)$: On the input of the ciphertext CT and a private key $PK_{\vec{v}}$, the algorithm begins to decrypt the ciphertext, but based on two conditions.

**Case I**: For a well-formed ciphertext, the algorithm decrypts $CT\;=\;\left(A,B,{\left\{C_{1,i},C_{2,i}\right\}}_{i=1}^{n},{\left\{C_{3,i},C_{4,i}\right\}}_{i=1}^{n},D=e{\left(g,g_{2}\right)}^{-s_{2}}M\right)$ using the private key $PK_{\vec{v}}=\left(K_{A},K_{B},{\left\{K_{1,i},K_{2,i}\right\}}_{i=1}^{n},{\left\{K_{3,i},K_{4,i}\right\}}_{i=1}^{n}\right)$ in order to output a message *M*, given by
$$M\leftarrow D\cdot e\left(A,K_{A}\right)\cdot e\left(B,K_{B}\right)\cdot \prod_{i=1}^{n}e(C_{1,i},K_{1,i}^{\prime })\cdot e(C_{2,i},K_{2,i}^{\prime })\cdot e(C_{3,i},K_{3,i}^{\prime })\cdot e(C_{4,i},K_{4,i}^{\prime })$$

**Correctness**: Assume the actual vector $\vec{x}=\left(x_{1},\ldots ,x_{n}\right)$ is used for the formation of the ciphertext CT. The message can be recovered as follows: Let $\beta =D\cdot e\left(A,K_{A}\right)\cdot e\left(B,K_{B}\right)$ and $\gamma =\prod_{i=1}^{n}e\left(C_{1,i},K_{1,i}\right)\cdot e\left(C_{2,i},K_{2,i}\right)\cdot e\left(C_{3,i},K_{3,i}\right)\cdot e\left(C_{4,i},K_{4,i}\right)$

Solving for γ, we have
$$\gamma =\prod_{i=1}^{n}e\left(g^{w_{1,i}s_{1}}g^{f_{1,i}s_{2}}g^{\delta_{1}x_{i}s_{3}},g^{-\delta_{2}r_{i}}g^{\lambda_{1}v_{i}w_{2,i}}\right)\cdot e\left(g^{w_{2,i}s_{1}}g^{f_{2,i}s_{2}}g^{\delta_{2}x_{i}s_{3}},g^{\delta_{1}r_{i}}g^{-\lambda_{1}v_{i}w_{1,i}}\right)$$

$$\cdot e\left(g^{t_{1,i}s_{1}}g^{h_{1,i}s_{2}}g^{\theta_{1}x_{i}s_{4}},g^{-\theta_{2}\phi_{i}}g^{\lambda_{2}v_{i}t_{2,i}}\right)\cdot e\left(g^{t_{2,i}s_{1}}g^{h_{2,i}s_{2}}g^{\theta_{2}x_{i}s_{4}},g^{\theta_{1}\phi_{i}}g^{-\lambda_{2}v_{i}t_{1,i}}\right)$$

$$=\prod_{i=1}^{n}e\left(g^{w_{1,i}s_{1}},g^{-\delta_{2}r_{i}}\right)\cdot e\left(g^{f_{1,i}s_{2}},g^{-\delta_{2}r_{i}}g^{\lambda_{1}v_{i}w_{2,i}}\right)\cdot e\left(g^{\delta_{1}x_{i}s_{3}},g^{\lambda_{1}v_{i}w_{2,i}}\right)\cdot e\left(g^{w_{2,i}s_{1}},g^{\delta_{1}r_{i}}\right)\cdot e\left(g^{f_{2,i}s_{2}},g^{\delta_{1}r_{i}}g^{-\lambda_{1}v_{i}w_{1,i}}\right)$$

$$\cdot e\left(g^{\delta_{2}x_{i}s_{3}},g^{-\lambda_{1}v_{i}w_{1,i}}\right)\cdot e\left(g^{t_{1,i}s_{1}},g^{-\theta_{2}\phi_{i}}\right)\cdot e\left(g^{h_{1,i}s_{2}},g^{-\theta_{2}\phi_{i}}g^{\lambda_{2}v_{i}t_{2,i}}\right)\cdot e\left(g^{\theta_{1}x_{i}s_{4}},g^{\lambda_{2}v_{i}t_{2,i}}\right)\cdot e\left(g^{t_{2,i}s_{1}},g^{\theta_{1}\phi_{i}}\right)$$

$$\cdot e\left(g^{h_{2,i}s_{2}},g^{\theta_{1}\phi_{i}}g^{-\lambda_{2}v_{i}t_{1,i}}\right)\cdot e\left(g^{\theta_{2}x_{i}s_{4}},g^{-\lambda_{2}v_{i}t_{1,i}}\right)$$

$$=\prod_{i=1}^{n}e\left(g^{-\delta_{2}w_{1,i}},g^{s_{1}r_{i}}\right)\cdot e\left(g^{s_{2}},{\left(g^{-\delta_{2}r_{i}}g^{\lambda_{1}v_{i}w_{2,i}}\right)}^{f_{1,i}}\right)\cdot e{\left(g,g\right)}^{\lambda_{1}\delta_{1}w_{2,1}\cdot x_{i}\cdot v_{i}\cdot s_{3}}\cdot e\left(g^{\delta_{1}w_{2,i}},g^{s_{1}r_{i}}\right)$$

$$\cdot e\left(g^{s_{2}},{\left(g^{\delta_{1}r_{i}}g^{-\lambda_{1}v_{i}w_{1,i}}\right)}^{f_{2,i}}\right)\cdot e{\left(g,g\right)}^{-\lambda_{1}\delta_{2}w_{1,1}\cdot x_{i}\cdot v_{i}\cdot s_{3}}\cdot e\left(g^{-\theta_{2}t_{1,i}},g^{s_{1}\phi_{i}}\right)\cdot e\left(g^{s_{2}},{\left(g^{-\theta_{2}\phi_{i}}g^{\lambda_{2}v_{i}t_{2,i}}\right)}^{h_{1,i}}\right)\cdot$$

$$e{\left(g,g\right)}^{\lambda_{2}\theta_{1}t_{2,1}\cdot x_{i}\cdot v_{i}\cdot s_{4}}\cdot e\left(g^{\theta_{1}t_{2,i}},g^{s_{1}\phi_{i}}\right)\cdot e\left(g^{s_{2}},{\left(g^{\theta_{1}\phi_{i}}g^{-\lambda_{2}v_{i}t_{1,i}}\right)}^{h_{2,i}}\right)\cdot e{\left(g,g\right)}^{-\lambda_{2}\theta_{2}t_{1,1}\cdot x_{i}\cdot v_{i}\cdot s_{4}}\cdot$$

$$=\prod_{i=1}^{n}e\left(g^{\delta_{1}w_{2,i}-\delta_{2}w_{1,i}},g^{r_{i}s_{1}}\right)\cdot \left(g^{\theta_{1}t_{2,i}-\theta_{2}t_{1,i}},g^{\phi_{i}s_{1}}\right)\cdot e\left(g^{s_{2}},K_{1,i}^{f_{1,i}}K_{2,i}^{f_{2,i}}K_{3,i}^{h_{1,i}}K_{4,i}^{h_{2,i}}\right)$$

$$\cdot e{\left(g,g\right)}^{\left[\lambda_{1}\left(\delta_{1}w_{2,i}-\delta_{2}w_{1,i}\right)s_{3}+\lambda_{2}\left(\theta_{1}t_{2,i}-\theta_{2}t_{1,i}\right)s_{4}\right]x_{i}\cdot v_{i}}$$

$$=\prod_{i=1}^{n}e\left(g^{\Psi },g^{r_{i}s_{1}}\right)\cdot e\left(g^{\Psi },g^{\phi_{i}s_{1}}\right)\cdot e\left(g^{s_{2}},K_{1,i}^{f_{1,i}}K_{2,i}^{f_{2,i}}K_{3,i}^{h_{1,i}}K_{4,i}^{h_{2,i}}\right)\cdot e{\left(g,g\right)}^{\Psi \left[\lambda_{1}s_{3}+\lambda_{2}s_{4}\right]\vec{x}\cdot \vec{v}}$$

$$=e\left(g^{\Psi s_{1}},\prod_{i=1}^{n}g^{\left(r_{i}+\phi_{i}\right)}\right)\cdot e\left(g^{s_{2}},\prod_{i=1}^{n}K_{1,i}^{f_{1,i}}K_{2,i}^{f_{2,i}}K_{3,i}^{h_{1,i}}K_{4,i}^{h_{2,i}}\right)\cdot e{\left(g,g\right)}^{\Psi \left[\lambda_{1}s_{3}+\lambda_{2}s_{4}\right]\vec{x}\cdot \vec{v}}$$

$$=e\left(g^{s_{2}},\prod_{i=1}^{n}g^{\left(r_{i}+\phi_{i}\right)}\right)\cdot e\left(g^{s_{2}},\prod_{i=1}^{n}K_{1,i}^{f_{1,i}}K_{2,i}^{f_{2,i}}K_{3,i}^{h_{1,i}}K_{4,i}^{h_{2,i}}\right)\cdot e{\left(g,g\right)}^{\Psi \left[\lambda_{1}s_{3}+\lambda_{2}s_{4}\right]\vec{x}\cdot \vec{v}}$$

The message *M* can then be recovered as,
$$M\leftarrow D\cdot e\left(A,K_{A}\right)\cdot e\left(B,K_{B}\right)\cdot \gamma$$

$$=e{\left(g,g_{2}\right)}^{-s_{2}}M\cdot e\left(g^{s_{2}},g_{2}\prod_{i=1}^{n}K_{1,i}^{-f_{1,i}}K_{2,i}^{-f_{2,i}}K_{3,i}^{-h_{1,i}}K_{4,i}^{-h_{2,i}}\right)\cdot e\left(g^{s_{2}},\prod_{i=1}^{n}g^{-\left(r_{i}+\phi_{i}\right)}\right)\cdot e\left(g^{s_{2}},\prod_{i=1}^{n}g^{\left(r_{i}+\phi_{i}\right)}\right)$$

$$\cdot e\left(g^{s_{2}},\prod_{i=1}^{n}K_{1,i}^{f_{1,i}}K_{2,i}^{f_{2,i}}K_{3,i}^{h_{1,i}}K_{4,i}^{h_{2,i}}\right)\cdot e{\left(g,g\right)}^{\Psi \left[\lambda_{1}s_{3}+\lambda_{2}s_{4}\right]\vec{x_{i}}\cdot \vec{v_{i}}}$$

$$=e{\left(g,g_{2}\right)}^{-s_{2}}M\cdot e{\left(g,g_{2}\right)}^{s_{2}}\cdot e{\left(g,g\right)}^{\Psi \left[\lambda_{1}s_{3}+\lambda_{2}s_{4}\right]\vec{x_{i}}\cdot \vec{v_{i}}}$$

$$=M\cdot e{\left(g,g\right)}^{\Psi \left[\lambda_{1}s_{3}+\lambda_{2}s_{4}\right]\vec{x_{i}}\cdot \vec{v_{i}}}$$

The above result outputs 1 iff $(\vec{x},\vec{v})=0$ in $Z_{p}$. If it happens that $(\vec{x},\vec{v})\neq 0$, then $\lambda_{1}s_{3}\;+\;\lambda_{2}s_{4}=0$. The probability of being the identity then becomes 1/p since the exponents $\lambda_{1}$, $\lambda_{2}$, $s_{3}$, and $s_{4}$ are all randomly chosen from $Z_{p}$.

**Case II**: However, for the re-encrypted version of the ciphertext, $CT\;=\;\left(A,\;B,\;CT_{A},CT_{B},D=e{\left(g,g_{2}\right)}^{-s_{2}}M\right)$, the algorithm first decrypts $CT_{A}$ by utilizing $PK_{\vec{v^{\prime }}}$, as shown below to obtain α. That is, α⟵$Decrypt(PK_{\vec{v^{\prime }}},CT_{A})$. The deduction and correctness are shown below. We first compute $CT_{A}$ as follows:$$CT_{A}=\prod_{i=1}^{n}\left\{g^{w_{1,i}s_{1}}g^{f_{1,i}s_{2}}g^{\delta_{1}x_{i}s_{3}},g^{w_{2,i}s_{2}}g^{f_{2,i}s_{2}}g^{\delta_{2}x_{i}s_{3}}\right\}\cdot \left\{g^{-\delta_{2}r_{i}^{\prime }}g^{\lambda_{i}^{\prime }v_{i}w_{2,i}}\alpha^{\delta_{2}},g^{\delta_{1}r_{i}^{\prime }}g^{-\lambda_{i}^{\prime }v_{i}w_{1,i}}\alpha^{-\delta_{1}}\right\}$$

$$=\prod_{i=1}^{n}e\left(g^{w_{1,i},s_{1}},g^{-\delta_{2}r_{i}^{\prime }}\right)\cdot e\left(g^{f_{1,i}s_{2}},g^{-\delta_{2},r_{i}^{\prime }}g^{\lambda_{1}^{\prime }v_{i}w_{2,i}}\alpha^{\delta_{2}}\right)\cdot e\left(g^{\delta_{1}x_{i}s_{3}},g^{\lambda_{1}^{\prime }v_{i}w_{2,i}}\right)\cdot e\left(g^{w_{1,i}s_{1}},\alpha^{s_{2}}\right)$$

$$\cdot e\left(g^{w_{2,i}s_{1}},g^{\delta_{1}r_{i}^{\prime }}\right)\cdot e\left(g^{f_{2,i}s_{2}},g^{\delta_{1},r_{i}^{\prime }}g^{-\lambda_{1}^{\prime }v_{i}w_{1,i}}\alpha^{-\delta_{1}}\right)\cdot e\left(g^{\delta_{2}x_{i}s_{3}},g^{-\lambda_{1}^{\prime }v_{i}w_{1,i}}\right)\cdot e\left(g^{w_{2,i}s_{2}},\alpha^{-\delta_{1}}\right)$$

$$=\prod_{i=1}^{n}e\left(g^{-\delta_{2}w_{1,i}},g^{r_{i}^{\prime }s_{1}}\right)\cdot e\left(g^{s_{2}},{\left(g^{-\delta_{2}r_{i}^{\prime }}g^{\lambda_{1}^{\prime }v_{i}w_{2,i}}\alpha^{\delta_{2}}\right)}^{f_{1,i}}\right)\cdot e{\left(g,g\right)}^{\lambda_{1}^{\prime }\delta_{1}w_{2,i}x_{i}v_{i}s_{3}}\cdot e\left(g^{w_{1,i}s_{1}},\alpha^{\delta_{2}}\right)$$

$$\cdot e\left(g^{\delta_{1}w_{2,i}},g^{r_{i}^{\prime }s_{1}}\right)\cdot e\left(g^{s_{2}},{\left(g^{\delta_{1}r_{i}^{\prime }}g^{-\lambda_{1}^{\prime }v_{i}w_{1,i}}\alpha^{-\delta_{1}}\right)}^{f_{2,i}}\right)\cdot e{\left(g,g\right)}^{-\lambda_{1}^{\prime }\delta_{2}w_{1,i}x_{i}v_{i}s_{3}}\cdot e\left(g^{w_{2,i}s_{2}},\alpha^{-\delta_{1}}\right)$$

$$=\prod_{i=1}^{n}e\left(g^{\delta_{1}w_{2,i}-\delta_{2}w_{1.i}},g^{r_{1}^{\prime }s_{1}}\right)\cdot e\left(g^{s_{2}},K_{1,i}^{{}^{\prime }f_{1,i}}K_{2,i}^{{}^{\prime }f_{2,i}}\right)\cdot e\left(g^{-\delta_{1}w_{2,i}+\delta_{2}w_{1,i}},\alpha^{s_{1}}\right)\cdot e{\left(g,g\right)}^{\lambda_{1}^{\prime }\left[\delta_{1}w_{2,i}-\delta_{2}w_{1,i}\right]x_{i}\cdot v_{i}\cdot s_{3}}$$

$$=\prod_{i=1}^{n}e\left(g^{\Psi },g^{r_{i}^{\prime }s_{1}}\right)\cdot e\left(g^{s_{2}},K_{1,i}^{{}^{\prime }f_{1,i}}K_{2,i}^{{}^{\prime }f_{2,i}}\right)\cdot e\left(g^{-\Psi },\alpha^{s_{1}}\right)\cdot e{\left(g,g\right)}^{\Psi \lambda_{1}^{\prime }x_{i}\cdot v_{i}\cdot s_{3}}$$

$$=e\left(g^{\Psi s_{1}},\prod_{i=1}^{n}g^{r_{i}^{\prime }s_{1}}\right)\cdot e\left(g^{s_{2}},\prod_{i=1}^{n}K_{1,i}^{{}^{\prime }f_{1,i}}K_{2,i}^{{}^{\prime }f_{2,i}}\right)\cdot e\left(g^{-\Psi s_{1}},\alpha \right)\cdot e{\left(g,g\right)}^{\Psi \lambda_{1}^{\prime }s_{3}\vec{x}\cdot \vec{v}}$$

$$=e\left(g^{s_{2}},\prod_{i=1}^{n}g^{r_{i}^{\prime }}\right)\cdot e\left(g^{s_{2}},\prod_{i=1}^{n}K_{1,i}^{{}^{\prime }f_{1,i}}K_{2,i}^{{}^{\prime }f_{2,i}}\right)\cdot e\left(g^{-s_{2}},\alpha \right)\cdot e{\left(g,g\right)}^{\Psi \lambda_{1}^{\prime }s_{3}\vec{x}\cdot \vec{v}}$$

Multiplying the result with $PK_{\vec{v^{\prime }}}$, we have
$$\alpha =\alpha \cdot e{\left(g,g_{2}\right)}^{\Psi \lambda_{1}^{\prime }s_{3}\left(\vec{x}\cdot \vec{v}\right)}$$

Thus, the output of the above is α iff $\vec{x}\cdot \vec{v}=0$. After completing the computation for α, we compute the message *M* as *M*⟵$D\cdot CT_{B}\cdot e\left(g^{\Psi s_{1}},\alpha \right)$, and it is shown below.
$$=e{\left(g,g_{2}\right)}^{-s_{2}}M\cdot e\left(g^{s_{2}},g_{2}\right)\cdot e{\left(g,g\right)}^{\Psi \left[\lambda_{1}^{\prime }s_{3}+\lambda_{2}^{\prime }s_{4}\right]\left(\vec{x}\cdot \vec{v}\right)}\cdot e\left(g^{-\Psi },\alpha^{s_{1}}\right)\cdot e\left(g^{\Psi s_{1}},\alpha \right)$$

Recalling that $\alpha =g_{2}^{l}$, we have
$$=e{\left(g,g_{2}\right)}^{-s_{2}}\cdot M\cdot e{\left(g,g_{2}\right)}^{s_{2}}\cdot e{\left(g,g\right)}^{\Psi \left[\lambda_{1}^{\prime }s_{3}+\lambda_{2}^{\prime }s_{4}\right]\left(\vec{x}\cdot \vec{v}\right)}\cdot e{\left(g,g_{2}\right)}^{-\Psi ls_{1}}\cdot e{\left(g,g_{2}\right)}^{\Psi ls_{1}}$$
$$=M\cdot e{\left(g,g\right)}^{\Psi \left[\lambda_{1}^{\prime }s_{3}+\lambda_{2}^{\prime }s_{4}\right]\left(\vec{x}\cdot \vec{v}\right)}$$

The above result outputs 1 iff $(\vec{x},\vec{v})=0$ in $Z_{p}$. If it happens that $(\vec{x},\vec{v})\neq 0$, then $\lambda_{1}^{\prime }s_{3}\;+\;\lambda_{2}^{\prime }s_{4}=0$. The probability of being the identity then becomes 1/p since the exponents are all randomly chosen from $Z_{p}$.

## 6. Security Model

Following the approach in [25], we prove that our scheme exhibits attribute-hiding property. The adversary, A, and the challenger, C, are engaged in a series of games in our security model. Both A and C are, by assumption, given the attribute set ∑, and the predicate class F beforehand. The security game is played over the vectors of the re-encryption process.

**Initialize**: The adversary, A, outputs two vectors $\vec{x^{\prime }},\vec{y^{\prime }}\in \sum$.

**Setup**: The challenger, C, runs Setup to obtain the public key $P_{pub}$ and the secret key SK, after which A is given $P_{pub}$.

**Query Phase 1**: A adaptively issues private key queries for the vector $\vec{v}=\left\{v_{i},\ldots ,v_{n}\right\}\in \sum$ subject to the restriction that, ∀i, $\langle \vec{v_{i}},\vec{x^{\prime }}\rangle =0$ iff $\langle \vec{v_{i}},\vec{y^{\prime }}\rangle =0$. C responds with $PK_{\vec{v}}\leftarrow \mathrm{KGen}\left(SK,\vec{v_{i}}\right)$

**Challenge**: Two messages $M_{0},M_{1}\in M$ are output by A. If $M_{0}\neq M_{1}$, it is a requirement that $\langle \vec{v_{i}},\vec{x^{\prime }}\rangle \neq \langle \vec{v_{i}},\vec{y^{\prime }}\rangle \neq 0$ for all queries made on the vector $\vec{v}$. C picks a random bit, b∈{0,1}. If b=0, C gives $CT^{\prime }\leftarrow \mathrm{Encrypt}\left(P_{pub},\vec{x^{\prime }},M_{0}\right)$ to A, and if b=1, $CT^{\prime }\leftarrow \mathrm{Encrypt}\left(P_{pub},\vec{y^{\prime }},M_{1}\right)$ is given to A.

**Query Phase 2**: Additional private key queries are made by A for additional vectors, subject to the same restrictions as stated above.

**Guess**: A outputs a guess $b^{\prime }\in \left\{0,1\right\}$, and wins the game if $b=b^{\prime }$.

The advantage of A is defined as $\mathrm{Adv}\left(A\right)=|Pr\left[b=b^{\prime }\right]-\frac{1}{2}|$. For the scenario where the two messages are not the same, i.e., $M_{0}\neq M_{1}$, A is not permitted to issue private key queries for vectors $\vec{v_{i}}$ such that $\langle \vec{v_{i}},\vec{x^{\prime }}\rangle =\langle \vec{v_{i}},\vec{y^{\prime }}\rangle =0$. This is done throughout all the query phases. If that is not the case, for a vector $\vec{v_{i}}$, the adversary can obtain a private key $PK_{\vec{v_{i}}}$ and decrypt the challenge ciphertext using the private key corresponding to that vector. The restriction is however not required for the case where $M_{0}=M_{1}$.

### Security Proof

In proving the security of our scheme, we introduce a series of security games between the adversary and the challenger as stated above. We also consider the case where there is a distinction between the two messages. As stated in the security model, the adversary is not in any way permitted to make private key queries for the vector $\vec{v}$ such that $\langle \vec{v_{i}},\vec{x^{\prime }}\rangle =\langle \vec{v_{i}},\vec{y^{\prime }}\rangle =0$.

$Game_{1}:$ The challenge ciphertext is generated under $\left(\vec{x^{\prime }},\vec{x^{\prime }}\right)$ and $M_{0}$, and it is computed as
$$CT_{1}=\left(g^{s_{2}},g_{1}^{s_{1}},{\left\{W_{1,i}^{s_{1}}\cdot F_{1,i}^{s_{2}}\cdot Q_{1}^{x_{i}^{\prime }s_{3}},W_{2,i}^{s_{1}}\cdot F_{2,i}^{s_{2}}\cdot Q_{2}^{x_{i}^{\prime }s_{3}}\right\}}_{i=1}^{n},{\left\{T_{1,i}^{s_{1}}\cdot H_{1,i}^{s_{2}}\cdot R_{1}^{x_{i}^{\prime }s_{4}},T_{2,i}^{s_{1}}\cdot H_{2,i}^{s_{2}}\cdot R_{2}^{x_{i}^{\prime }s_{4}}\right\}}_{i=1}^{n},\Upsilon^{-s_{2}}M_{0}\right)$$

$Game_{2}:$$\left(\vec{x^{\prime }},\vec{x^{\prime }}\right)$ and a random message $R_{x}\in G_{T}$ are used to generate the challenge ciphertext, and it is computed as
$$CT_{2}=\left(g^{s_{2}},g_{1}^{s_{1}},{\left\{W_{1,i}^{s_{1}}\cdot F_{1,i}^{s_{2}}\cdot Q_{1}^{x_{i}^{\prime }s_{3}},W_{2,i}^{s_{1}}\cdot F_{2,i}^{s_{2}}\cdot Q_{2}^{x_{i}^{\prime }s_{3}}\right\}}_{i=1}^{n},{\left\{T_{1,i}^{s_{1}}\cdot H_{1,i}^{s_{2}}\cdot R_{1}^{x_{i}^{\prime }s_{4}},T_{2,i}^{s_{1}}\cdot H_{2,i}^{s_{2}}\cdot R_{2}^{x_{i}^{\prime }s_{4}}\right\}}_{i=1}^{n},R_{x}\right)$$

$Game_{3}:$ The challenge ciphertext is generated under $\left(\vec{x^{\prime }},\vec{0}\right)$ and a random message $R_{x}\in G_{T}$, and it is computed as
$$CT_{3}=\left(g^{s_{2}},g_{1}^{s_{1}},{\left\{W_{1,i}^{s_{1}}\cdot F_{1,i}^{s_{2}}\cdot Q_{1}^{x_{i}^{\prime }s_{3}},W_{2,i}^{s_{1}}\cdot F_{2,i}^{s_{2}}\cdot Q_{2}^{x_{i}^{\prime }s_{3}}\right\}}_{i=1}^{n},{\left\{T_{1,i}^{s_{1}}\cdot H_{1,i}^{s_{2}},T_{2,i}^{s_{1}}\cdot H_{2,i}^{s_{2}}\right\}}_{i=1}^{n},R_{x}\right)$$

$Game_{4}:$ The challenge ciphertext is generated under $\left(\vec{x^{\prime }},\vec{y^{\prime }}\right)$ and a random message $R_{x}\in G_{T}$, and it is computed as
$$CT_{4}=\left(g^{s_{2}},g_{1}^{s_{1}},{\left\{W_{1,i}^{s_{1}}\cdot F_{1,i}^{s_{2}}\cdot Q_{1}^{x_{i}^{\prime }s_{3}},W_{2,i}^{s_{1}}\cdot F_{2,i}^{s_{2}}\cdot Q_{2}^{x_{i}^{\prime }s_{3}}\right\}}_{i=1}^{n},{\left\{T_{1,i}^{s_{1}}\cdot H_{1,i}^{s_{2}}\cdot R_{1}^{y_{i}^{\prime }s_{4}},T_{2,i}^{s_{1}}\cdot H_{2,i}^{s_{2}}\cdot R_{2}^{y_{i}^{\prime }s_{4}}\right\}}_{i=1}^{n},R_{x}\right)$$

$Game_{5}:$ The challenge ciphertext is generated under $\left(\vec{0},\vec{y^{\prime }}\right)$ and a random message $R_{x}\in G_{T}$, and it is computed as
$$CT_{5}=\left(g^{s_{2}},g_{1}^{s_{1}},{\left\{W_{1,i}^{s_{1}}\cdot F_{1,i}^{s_{2}},W_{2,i}^{s_{1}}\cdot F_{2,i}^{s_{2}}\right\}}_{i=1}^{n},{\left\{T_{1,i}^{s_{1}}\cdot H_{1,i}^{s_{2}}\cdot R_{1}^{y_{i}^{\prime }s_{4}},T_{2,i}^{s_{1}}\cdot H_{2,i}^{s_{2}}\cdot R_{2}^{y_{i}^{\prime }s_{4}}\right\}}_{i=1}^{n},R_{x}\right)$$

$Game_{6}:$ The challenge ciphertext is generated under $\left(\vec{y^{\prime }},\vec{y^{\prime }}\right)$ and a random message $R_{x}\in G_{T}$, and it is computed as
$$CT_{6}=\left(g^{s_{2}},g_{1}^{s_{1}},{\left\{W_{1,i}^{s_{1}}\cdot F_{1,i}^{s_{2}}\cdot Q_{1}^{y_{i}^{\prime }s_{3}},W_{2,i}^{s_{1}}\cdot F_{2,i}^{s_{2}}\cdot Q_{2}^{y_{i}^{\prime }s_{3}}\right\}}_{i=1}^{n},{\left\{T_{1,i}^{s_{1}}\cdot H_{1,i}^{s_{2}}\cdot R_{1}^{y_{i}^{\prime }s_{4}},T_{2,i}^{s_{1}}\cdot H_{2,i}^{s_{2}}\cdot R_{2}^{y_{i}^{\prime }s_{4}}\right\}}_{i=1}^{n},R_{x}\right)$$

$Game_{7}:$$\left(\vec{y^{\prime }},\vec{y^{\prime }}\right)$ and $M_{1}$ is used to generate the challenge ciphertext, and it is computed as
$$CT_{7}=\left(g^{s_{2}},g_{1}^{s_{1}},{\left\{W_{1,i}^{s_{1}}\cdot F_{1,i}^{s_{2}}\cdot Q_{1}^{y_{i}^{\prime }s_{3}},W_{2,i}^{s_{1}}\cdot F_{2,i}^{s_{2}}\cdot Q_{2}^{y_{i}^{\prime }s_{3}}\right\}}_{i=1}^{n},{\left\{T_{1,i}^{s_{1}}\cdot H_{1,i}^{s_{2}}\cdot R_{1}^{y_{i}^{\prime }s_{4}},T_{2,i}^{s_{1}}\cdot H_{2,i}^{s_{2}}\cdot R_{2}^{y_{i}^{\prime }s_{4}}\right\}}_{i=1}^{n},\Upsilon^{-s_{2}}M_{1}\right)$$

We prove that $Game_{1}$ and $Game_{7}$ are indistinguishable to an adversary with polynomial time. This is achieved by proving the computational indistinguishability of the transitions between the games. This is because the indistinguishability between $Game_{1}$ and $Game_{2}$ also indicates that $Game_{6}$ and $Game_{7}$ are also indistinguishable, by the property of symmetry of the hybrid games [25].

Under the (t,ϵ) Decision Bilinear Diffie–Hellman assumption, $Game_{1}$ and $Game_{2}$ cannot be distinguished by an adversary running in polynomial time *t* with an advantage greater than ϵ, assuming there is an adversary A with non-negligible advantage ϵ that can attack the scheme. We describe the game between the challenger and the adversary as follows. On input $\left(g,g^{a},g^{b},g^{c},Z\right)\in \;G^{4}\;\times \;G_{T}$, the goal of the challenger is to output 1 if $Z=g^{abc}$, and 0 otherwise. The challenger and the adversary engage in the following interaction:

**Public parameters:** The challenger chooses random exponents ${\left\{\delta_{i},\theta_{i}\right\}}_{i=1}^{2}$, ${\left\{w_{1,i},t_{1,i}\right\}}_{i=1}^{n}$, ${\left\{f_{1,i},f_{2,i}\right\}}_{i=1}^{n}$, ${\left\{h_{1,i},h_{2,i}\right\}}_{i=1}^{n},and\;\omega \in Z_{p}$. A random $\Psi \in Z_{p}$ is also selected to obtain ${\left\{w_{2,i},t_{2,i}\right\}}_{i=1}^{n}$ under the constraints
$$\Psi =\delta_{1}w_{2,i}-\delta_{2}w_{1,i},\Psi =\theta_{1}t_{2,i}-\theta_{2}t_{1,i}$$

If Ψ=0, the challenger selects a new set of random exponents. It then sets the following conditions
$$W_{1,i}=g^{w_{1,i}},W_{2,i}=g^{w_{2,i}},T_{1,i}=g^{t_{1,i}},F_{1,i}=g^{\hat{f_{1,i}}},F_{2,i}=g^{\hat{f_{2,i}}},H_{1,i}=g^{\hat{h_{1,i}}},H_{2,i}=g^{\hat{h_{2,i}}}T_{2,i}=g^{t_{2,i}}$$
where
$$\hat{f_{1,i}}=x_{i}^{\prime }\delta_{1}b+f_{1,i},\hat{f_{2,i}}=x_{i}^{\prime }\delta_{2}b+f_{2,i},\hat{h_{1,i}}=x_{i}^{\prime }\theta_{1}b+h_{1,i},\hat{h_{2,i}}=x_{i}^{\prime }\theta_{2}b+h_{2,i}$$
∀i=1,…,n and $g_{2}=g^{-\Psi ab}g^{\omega }$.

The challenger then initiates the following notations:$$Q_{1}=g^{\delta_{1}},Q_{2}=g^{\delta_{2}},R_{1}=g^{\theta_{1}},R_{2}=g^{\theta_{2}},g_{1}=g^{\Psi },\Upsilon =e{\left(g^{a},g^{b}\right)}^{-\Psi }\cdot e{\left(g,g\right)}^{\omega }$$

**Key Derivation:**A issues private key queries for the vectors. Considering making queries for the vector $\vec{v}=\left(v_{1},\ldots ,v_{n}\right)\in Z_{p}$, A can request for private key queries as long as $\langle \vec{v},\vec{x^{\prime }}\rangle =\varrho \neq 0$. The challenger selects random exponents $\lambda_{1}^{\prime },\lambda_{2}^{\prime }$, ${\left\{r_{i}^{\prime },\phi_{i}^{\prime }\right\}}_{i=1}^{n}\in Z_{p}$ in generating the re-encrypted key $REKey_{\vec{v}}$, and sets
$$\hat{\lambda_{1}^{\prime }}=\mu a+\lambda_{1}^{\prime },\hat{\lambda_{2}^{\prime }}=\mu a+\lambda_{2}^{\prime }$$
where $\mu =\frac{1}{2\varrho }$. The re-encrypted keys $K_{1,i}^{\prime },K_{2,i}^{\prime },K_{3,i}^{\prime },K_{4,i}^{\prime }$ are then generated as follows:$$K_{1,i}^{\prime }={\left(g^{a}\right)}^{v_{i}w_{2,i}\mu }g^{-\delta_{2}r_{i}^{\prime }}g^{\lambda_{1}^{\prime }v_{i}w_{2,i}}\alpha^{\delta_{2}},K_{2,i}^{\prime }={\left(g^{a}\right)}^{-v_{i}w_{1,i}\mu }g^{\delta_{1}r_{i}^{\prime }}g^{-\lambda_{1}^{\prime }v_{i}w_{1,i}}\alpha^{-\delta_{1}},$$
$$K_{3,i}^{\prime }={\left(g^{a}\right)}^{v_{i}t_{2,i}\mu }g^{-\theta_{1}\phi_{i}^{\prime }}g^{\lambda_{2}^{\prime }v_{i}t_{2,i}}\alpha^{\theta_{2}},K_{4,i}^{\prime }={\left(g^{a}\right)}^{-v_{i}t_{1,i}\mu }g^{\theta_{1}\phi_{i}^{\prime }}g^{-\lambda_{2}^{\prime }v_{i}t_{1,i}}\alpha^{-\theta_{1}},$$
∀i=1,…,n. The $K_{A}^{\prime }$ and $K_{B}^{\prime }$ elements are, respectively, computed as $K_{A}^{\prime }\;=\;g_{2}\prod_{i=1}^{n}K_{1,i}^{{}^{\prime }-\hat{f_{1,i}}}K_{2,i}^{{}^{\prime }\hat{-f_{2,i}}}K_{3,i}^{{}^{\prime }-\hat{h_{1,i}}}K_{4,i}^{{}^{\prime }\hat{-h_{2,i}}}$ and $K_{B}^{\prime }=\prod_{i=1}^{n}g^{-(r_{i}^{\prime }+\phi_{i}^{\prime })}$.

Let $X=K_{1,i}^{{}^{\prime }-\hat{f_{1,i}}}K_{2,i}^{{}^{\prime }\hat{-f_{2,i}}}$ and $Y=K_{3,i}^{{}^{\prime }-\hat{h_{1,i}}}K_{4,i}^{{}^{\prime }\hat{-h_{2,i}}}$. Computing for both X and Y yields
$$X={\left[{\left(g^{a}\right)}^{v_{i}w_{2,i}\mu }g^{-\delta_{2}r_{i}^{\prime }}g^{\lambda_{1}^{\prime }v_{i}w_{2,i}}\alpha^{\delta_{2}}\right]}^{-\left(x_{i}^{\prime }\delta_{1}b+f_{1,i}\right)}\cdot {\left[{\left(g^{a}\right)}^{-v_{i}w_{1,i}\mu }g^{\delta_{1}r_{i}^{\prime }}g^{-\lambda_{1}^{\prime }v_{i}w_{1,i}}\alpha^{-\delta_{1}}\right]}^{-\left(x_{i}^{\prime }\delta_{2}b+f_{2,i}\right)}$$
$$={\left(g^{ab}\right)}^{-v_{i}x_{i}^{\prime }\delta_{1}w_{2,i}\mu }{\left(g^{a}\right)}^{-v_{i}w_{2,i}f_{1,i}\mu }{\left(g^{b}\right)}^{x_{i}^{\prime }\delta_{1}\delta_{2}r_{i}^{\prime }}g^{r_{i}^{\prime }\delta_{2}f_{1,i}}{\left(g^{b}\right)}^{-\lambda_{1}^{\prime }v_{i}x_{i}^{\prime }\delta_{1}w_{2,i}}g^{-\lambda_{1}^{\prime }v_{i}w_{2,i}f_{1,i}}{\left(\alpha^{b}\right)}^{-x_{i}^{\prime }\delta_{1}\delta_{2}}\alpha^{-\delta_{2}f_{1,i}}$$
$$\cdot {\left(g^{ab}\right)}^{v_{i}x_{i}^{\prime }\delta_{2}w_{1,i}\mu }{\left(g^{a}\right)}^{v_{i}w_{1,i}f_{2,i}\mu }{\left(g^{b}\right)}^{-x_{i}^{\prime }\delta_{1}\delta_{2}r_{i}^{\prime }}g^{-r_{i}^{\prime }\delta_{1}f_{2,i}}{\left(g^{b}\right)}^{\lambda_{1}^{\prime }v_{i}x_{i}^{\prime }\delta_{2}w_{1,i}}g^{\lambda_{1}^{\prime }v_{i}w_{1,i}f_{2,i}}{\left(\alpha^{b}\right)}^{x_{i}^{\prime }\delta_{1}\delta_{2}}\alpha^{\delta_{1}f_{2,i}}$$
$$={\left(g^{ab}\right)}^{v_{i}x_{i}^{\prime }\left[\delta_{2}w_{1,i}-\delta_{1}w_{2,i}\right]\mu }{\left(g^{a}\right)}^{v_{i}\left[w_{1,i}f_{2,i}-w_{2,i}f_{1,i}\right]\mu }g^{r_{i}^{\prime }\left[\delta_{2}f_{1,i}-\delta_{1}f_{2,i}\right]}{\left(g^{b}\right)}^{\lambda_{1}^{\prime }v_{i}x_{i}^{\prime }\left[\delta_{2}w_{1,i}-\delta_{1}w_{2,i}\right]}g^{\lambda_{1}^{\prime }v_{i}\left[w_{1,i}f_{2,i}-w_{2,i}f_{1,i}\right]}$$
$$\alpha^{\left[\delta_{1}f_{2,i}-\delta_{2}f_{1,i}\right]}$$
$$={\left(g^{ab}\right)}^{-\Psi v_{i}x_{i}^{\prime }\mu }{\left(g^{a}\right)}^{\chi v_{i}\mu }g^{r_{i}^{\prime }\vartheta }{\left(g^{b}\right)}^{-\Psi \lambda_{1}^{\prime }v_{i}x_{i}^{\prime }}g^{\chi \lambda_{1}^{\prime }v_{i}}\alpha^{-\vartheta }$$
where $\Psi =\delta_{1}w_{2,i}-\delta_{2}w_{1,i}$, $\chi =w_{1,i}f_{2,i}-w_{2,i}f_{1,i}$ and $\vartheta =\delta_{2}f_{1,i}-\delta_{1}f_{2,i}$.
$$Y={\left[{\left(g^{a}\right)}^{v_{i}t_{2,i}\mu }g^{-\theta_{2}\phi_{i}^{\prime }}g^{\lambda_{2}^{\prime }v_{i}t_{2,i}}\alpha^{\theta_{2}}\right]}^{-\left(x_{i}^{\prime }\theta_{1}b+h_{1,i}\right)}\cdot {\left[{\left(g^{a}\right)}^{-v_{i}t_{1,i}\mu }g^{\theta_{1}\phi_{i}^{\prime }}g^{-\lambda_{2}^{\prime }v_{i}t_{1,i}}\alpha^{-\theta_{1}}\right]}^{-\left(x_{i}^{\prime }\theta_{2}b+h_{2,i}\right)}$$
$$={\left(g^{ab}\right)}^{-v_{i}x_{i}^{\prime }\theta_{1}t_{2,i}\mu }{\left(g^{a}\right)}^{-v_{i}t_{2,i}h_{1,i}\mu }{\left(g^{b}\right)}^{x_{i}^{\prime }\theta_{1}\theta_{2}\phi_{i}^{\prime }}g^{\phi_{i}^{\prime }\theta_{2}h_{1,i}}{\left(g^{b}\right)}^{-\lambda_{2}^{\prime }v_{i}x_{i}^{\prime }\theta_{1}t_{2,i}}g^{-\lambda_{2}^{\prime }v_{i}t_{2,i}h_{1,i}}{\left(\alpha^{b}\right)}^{-x_{i}^{\prime }\theta_{1}\theta_{2}}\alpha^{-\theta_{2}h_{1,i}}$$
$$\cdot {\left(g^{ab}\right)}^{v_{i}x_{i}^{\prime }\theta_{2}t_{1,i}\mu }{\left(g^{a}\right)}^{v_{i}t_{1,i}h_{2,i}\mu }{\left(g^{b}\right)}^{-x_{i}^{\prime }\theta_{1}\theta_{2}\phi_{i}^{\prime }}g^{-\phi_{i}^{\prime }\theta_{1}h_{2,i}}{\left(g^{b}\right)}^{\lambda_{2}^{\prime }v_{i}x_{i}^{\prime }\theta_{2}t_{1,i}}g^{\lambda_{2}^{\prime }v_{i}t_{1,i}h_{2,i}}{\left(\alpha^{b}\right)}^{x_{i}^{\prime }\theta_{1}\theta_{2}}\alpha^{\theta_{1}h_{2,i}}$$
$$={\left(g^{ab}\right)}^{v_{i}x_{i}^{\prime }\left[\theta_{2}t_{1,i}-\theta_{1}t_{2,i}\right]\mu }{\left(g^{a}\right)}^{v_{i}\left[t_{1,i}h_{2,i}-t_{2,i}h_{1,i}\right]\mu }g^{\phi_{i}^{\prime }\left[\theta_{2}h_{1,i}-\theta_{1}h_{2,i}\right]}{\left(g^{b}\right)}^{\lambda_{2}^{\prime }v_{i}x_{i}^{\prime }\left[\theta_{2}t_{1,i}-\theta_{1}t_{2,i}\right]}g^{\lambda_{2}^{\prime }v_{i}\left[t_{1,i}h_{2,i}-t_{2,i}h_{1,i}\right]}$$
$$\alpha^{\left[\theta_{1}h_{2,i}-\theta_{2}h_{1,i}\right]}$$
$$={\left(g^{ab}\right)}^{-\Psi v_{i}x_{i}^{\prime }\mu }{\left(g^{a}\right)}^{\zeta v_{i}\mu }g^{\phi_{i}^{\prime }\xi }{\left(g^{b}\right)}^{-\Psi \lambda_{2}^{\prime }v_{i}x_{i}^{\prime }}g^{\zeta \lambda_{2}^{\prime }v_{i}}\alpha^{-\xi }$$
where $\Psi =\theta_{1}t_{2,i}-\theta_{2}t_{1,i}$, $\zeta =t_{1,i}h_{2,i}-t_{2,i}h_{1,i}$ and $\xi =\theta_{2}h_{1,i}-\theta_{1}h_{2,i}$.

X·Y results in
$$X\cdot Y={\left(g^{ab}\right)}^{-2\Psi v_{i}x_{i}^{\prime }\mu }{\left(g^{a}\right)}^{v_{i}\left[\chi +\zeta \right]\mu }g^{r_{i}^{\prime }\vartheta +\phi_{i}^{\prime }\xi }{\left(g^{b}\right)}^{-\Psi v_{i}x_{i}^{\prime }\left[\lambda_{1}^{\prime }+\lambda_{2}^{\prime }\right]}g^{v_{i}\left[\chi \lambda_{1}^{\prime }+\zeta \lambda_{2}^{\prime }\right]}\alpha^{-\left(\vartheta +\xi \right)}$$

The challenger can then compute $K_{A}^{\prime }$ as
$$K_{A}^{\prime }=g_{2}\prod_{i=1}^{n}{\left(g^{ab}\right)}^{-2\Psi v_{i}x_{i}^{\prime }\mu }{\left(g^{a}\right)}^{v_{i}\left[\chi +\zeta \right]\mu }{\left(g^{b}\right)}^{-\Psi v_{i}x_{i}^{\prime }\left[\lambda_{1}^{\prime }+\lambda_{2}^{\prime }\right]}\cdot g^{r_{i}^{\prime }\vartheta +\phi_{i}^{\prime }\xi +v_{i}\left[\chi \lambda_{1}^{\prime }+\zeta \lambda_{2}^{\prime }\right]}\alpha^{-\left(\vartheta +\xi \right)}$$
$$=g^{\omega }\prod_{i=1}^{n}{\left(g^{a}\right)}^{v_{i}\left[\chi +\zeta \right]\mu }\cdot \prod_{i=1}^{n}{\left(g^{b}\right)}^{-\Psi v_{i}x_{i}^{\prime }\left[\lambda_{1}^{\prime }+\lambda_{2}^{\prime }\right]}\cdot \prod_{i=1}^{n}g^{r_{i}^{\prime }\vartheta +\phi_{i}^{\prime }\xi +v_{i}\left[\chi \lambda_{1}^{\prime }+\zeta \lambda_{2}^{\prime }\right]}\cdot \alpha^{-\left(\vartheta +\xi \right)}$$

The challenger issues the private key $PK_{\vec{v^{\prime }}}=\left(K_{A}^{\prime },K_{B}^{\prime },{\left\{K_{1,i}^{\prime },K_{2,i}^{\prime }\right\}}_{i=1}^{n},{\left\{K_{3,i}^{\prime },K_{4,i}^{\prime }\right\}}_{i=1}^{n}\right)$ for the queried vector.

**Challenge Ciphertext:** In generating the challenge ciphertext, the challenger selects random elements $s_{1},s_{3},s_{4}\in Z_{p}$, and sets
$$\hat{s_{1}}=s_{1},\hat{s_{2}}=c,\hat{s_{3}}=s_{3}-bc,\hat{s_{4}}=s_{4}-bc.$$

The challenger then computes $A=g^{s_{2}}=g^{c}$ and $B=g^{\Psi s_{1}}={\left(g^{\Psi }\right)}^{s_{1}}=g_{1}^{s_{1}}$, and ∀i=1,…,n, the ciphertexts $C_{1,i},C_{2,i},C_{3,i},C_{4,i}$ are computed as follows
$$C_{1,i}=g^{w_{1,i}s_{1}}\cdot {\left(g^{c}\right)}^{\hat{f_{1,i}}}\cdot g^{\delta_{1}x_{i}^{\prime }s_{3}}={\left(g^{w_{1,i}}\right)}^{s_{1}}\cdot {\left({\left(g^{b}\right)}^{x_{i}^{\prime }\delta_{1}}g^{f_{1,i}}\right)}^{c}\cdot {\left(g^{\delta_{1}}\right)}^{x_{i}^{\prime }\left(s_{3}-bc\right)}=W_{1,i}^{\hat{s_{1}}}\cdot F_{1,i}^{\hat{s_{2}}}\cdot Q_{1}^{x_{i}^{\prime }\hat{s_{3}}}$$

$$C_{2,i}=g^{w_{2,i}s_{1}}\cdot {\left(g^{c}\right)}^{\hat{f_{2,i}}}\cdot g^{\delta_{2}x_{i}^{\prime }s_{3}}={\left(g^{w_{2,i}}\right)}^{s_{1}}\cdot {\left({\left(g^{b}\right)}^{x_{i}^{\prime }\delta_{2}}g^{f_{2,i}}\right)}^{c}\cdot {\left(g^{\delta_{2}}\right)}^{x_{i}^{\prime }\left(s_{3}-bc\right)}=W_{2,i}^{\hat{s_{1}}}\cdot F_{2,i}^{\hat{s_{2}}}\cdot Q_{2}^{x_{i}^{\prime }\hat{s_{3}}}$$

$$C_{3,i}=g^{t_{1,i}s_{1}}\cdot {\left(g^{c}\right)}^{\hat{h_{1,i}}}\cdot g^{\theta_{1}x_{i}^{\prime }s_{4}}={\left(g^{t_{1,i}}\right)}^{s_{1}}\cdot {\left({\left(g^{b}\right)}^{x_{i}^{\prime }\theta_{1}}g^{h_{1,i}}\right)}^{c}\cdot {\left(g^{\theta_{1}}\right)}^{x_{i}^{\prime }\left(s_{4}-bc\right)}=T_{1,i}^{\hat{s_{1}}}\cdot H_{1,i}^{\hat{s_{2}}}\cdot R_{1}^{x_{i}^{\prime }\hat{s_{4}}}$$

$$C_{4,i}=g^{t_{2,i}s_{1}}\cdot {\left(g^{c}\right)}^{\hat{h_{2,i}}}\cdot g^{\theta_{2}x_{i}^{\prime }s_{4}}={\left(g^{t_{2,i}}\right)}^{s_{1}}\cdot {\left({\left(g^{b}\right)}^{x_{i}^{\prime }\theta_{2}}g^{h_{2,i}}\right)}^{c}\cdot {\left(g^{\theta_{2}}\right)}^{x_{i}^{\prime }\left(s_{4}-bc\right)}=T_{2,i}^{\hat{s_{1}}}\cdot H_{2,i}^{\hat{s_{2}}}\cdot R_{2}^{x_{i}^{\prime }\hat{s_{4}}}$$

The challenger then computes $D=Z^{-\Psi }\cdot e{\left(g,g^{c}\right)}^{\omega }\cdot M_{0}$.

Under the Decisional BDH assumption, $Game_{1}$ and $Game_{2}$ are indistinguishable since, if $Z\;=\;e{\left(g,g\right)}^{abc}$, the challenge ciphertext is as given in $Game_{1}$, while, if *Z* is a randomly chosen element in $G_{T}$, then the challenge ciphertext is as shown in $Game_{2}$.

## 7. Implementation and Performance Analysis

In this section, we provide details of the implementation of our system and also evaluate the performance of our system. Experiments were designed and some useful parameters were measured. In our system, users (data owners inclusive) are registered on the blockchain network and this involves aggregating information pertaining to a specific user. Users are categorized as specified by the data owner. Each user is then given a public and private key pair, which are associated with their details, and to be used in requesting and accessing data.

We implemented the blockchain system on a private Ethereum blockchain network. Ethereum is a programmable blockchain platform that utilizes the robust nature of Solidity (a state-based scripting language). An application was designed in Python that connects each data owner and performs the proxy re-encryption scheme on the data. This application synchronizes with the blockchain using the JSON-RPC (JavaScript Object Notation—Remote Procedure Calls) library. With the blockchain notified about data request, queries are sent to the cloud server and data are filtered and sent to the blockchain. Re-encryption is either performed or not, based on the user type.

### 7.1. Experiment 1

In this first experiment, we measured the time it takes to register a user (both data owner and data user) on the blockchain network. To register, the user sends its details to the blockchain and membership keys are given to the user. We measured the delay it takes in mining this transaction. Variations over 40 runs of this scenario were simulated and the average registration delay was obtained. Experiment results indicate an average delay of 13.94 s, which is not far off the 13 s for a block generation in Ethereum networks. The experiment result is shown in Figure 2.

### 7.2. Experiment 2

In this second experiment, the impact of proxy re-encryption was measured. A flow chart, as shown in Figure 3, was designed that describes data processing as the data are requested by a user. As soon as data request is made, the blockchain network checks if the user is a legitimate member of the network. If successful, it sends a notification to the cloud server, which then filters and retrieves the data before sending them back to the blockchain network.

After receiving the data, the blockchain checks for the user type. For a primary user, the blockchain delivers the data and proceeds to mine the address and this becomes a transaction. For a secondary user, the proxy is called upon and it re-encrypts the data before giving it out, after which it is also mined. Experiment results are shown in Figure 4.

The tests were run for a variation of 40 times and it was realized that it takes an average of 30.18 s for an end-to-end data processing without re-encryption (as described in the flow chart) to be completed. Similarly, an average of 47.73 s was recorded for a process that involves re-encryption. Consequently, we realized the addition of re-encryption to the scheme increased the delay by 58.15%.

## 8. Discussion

**Collusion Resistance:** Our proposed scheme prevents collusion attack in the sense that the re-encrypted data are divided into two parts with one part stored on the blockchain network, and the other part stored on the cloud. Because the blockchain network and the cloud server work in tandem, a data user has to first obtain the bit-part data stored on the blockchain before obtaining the other half from the cloud. As a first level security check (usually performed before decryption), a data user must prove to the blockchain networks’ verification unit its membership before gaining access to the data. A revoked user is deprived of this right because its membership keys have been completely removed from the network and therefore the user becomes unknown to the network.However, for a revoked user who still colludes with the cloud server for access to data, the cloud server still has to provide the user’s details to the blockchain processing node for the necessary checks to be made. With collusion attack prevented, the confidentiality of the data is preserved/guaranteed.**Fine-grained access control:** There is an effective management of user access by the implementation of the ABE scheme. The utilization of the inner product encryption scheme enables a fine-grained access control to data. The data owner specifies which attribute set or right a data user enjoys and therefore, to access data, there should be a match-up between the attribute set and the private key set. There is also the possibility of selective delegation due to the weight (information type) set by the data owner. Furthermore, depending on the level of trust between the data owner and the user(s), decryption of either all or some data can be delegated selectively to the user(s).

## 9. Conclusions

In this paper, an inner-product proxy re-encryption scheme that ensures an efficient and secured data access to IoT data is presented. The encryption of IoT data is done according to a given access policy and shared with the various data users, and therefore the problem of data sharing has been addressed. We incorporated a blockchain network, whose processing node acts as the proxy server. A user can access data when it is a registered member of the network, with the verification performed by the blockchain network. The proxy also re-encrypts the data by transforming the policy set in the process of sharing the data. The blockchain network works in tandem with the cloud server to ensure a collusion-resistant scheme. Our approach also achieves a fine-grained access control to data. Experiment results show that proxy re-encryption increased the delay, but the utilization of a blockchain kept a record of all interactions between entities and eliminated the need of a trusted third party. Making improvements to our scheme, in terms of its efficiency, is the focus of our future work. We also plan to include a detailed smart contract algorithm and more experimental results in the next work.

## Figures and Tables

**Figure 1 sensors-19-01235-f001:**
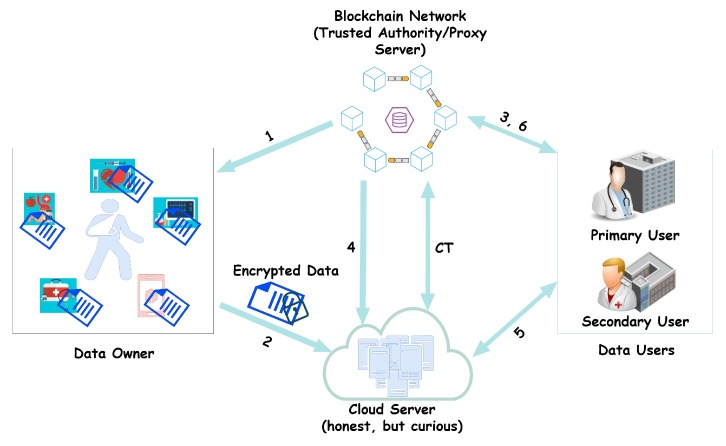
System model.

**Figure 2 sensors-19-01235-f002:**
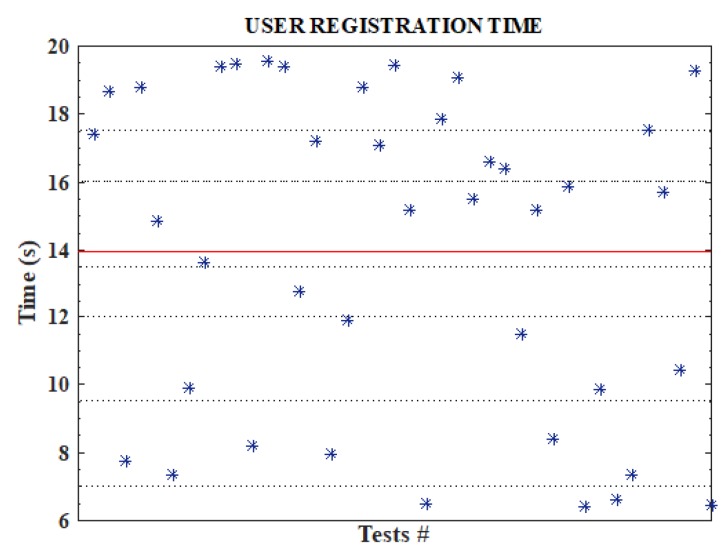
User registration delay.

**Figure 3 sensors-19-01235-f003:**
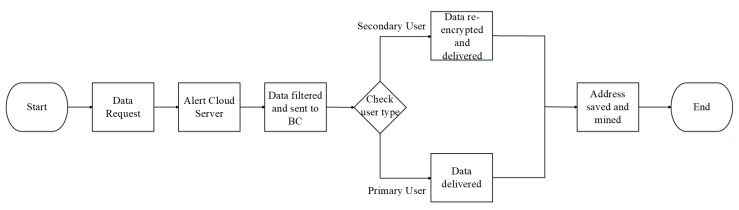
Flow chart.

**Figure 4 sensors-19-01235-f004:**
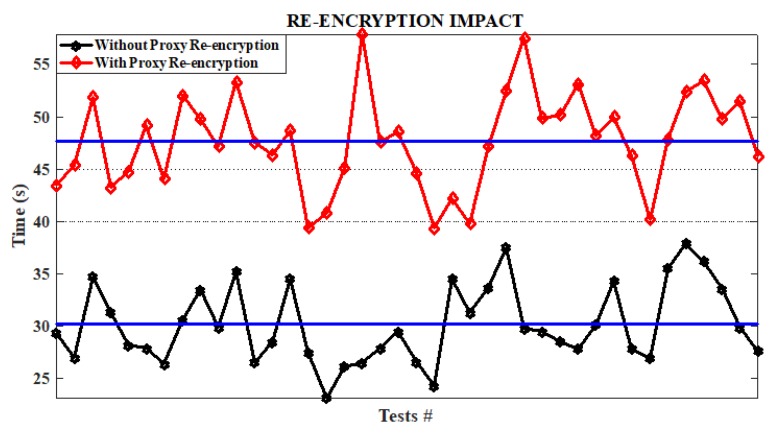
Impact of proxy re-encryption.

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
