# Peer review of "A Secured Proxy-Based Data Sharing Module in IoT Environments Using Blockchain"

_sensors, 2019, doi:10.3390/s19051235_

Round 1
Reviewer 1 Report
The paper significantly contributes to the body of knowledge by clearly presenting issues and solutions related to the sharing of information in IoT environments. The solution presented for the specified issue is adequately addressed by the use of Proxy Re-encryption and blockchain with emphasis on Inner Product Encryption. The references presented serve as verification of research on the previously mentioned issue. Descriptions on processes of the proposed solution backed by background definitions on the proposed solution are well outlined. The security scheme, model and proofs have been carefully constructed. However, the following minor changes are required.
- All figures must be referred in the manuscript. Figure 1 was not referred in the manuscript
- Review the entire manuscript and rewrite parts of it.
Author Response
Dear Reviewer, thank you for your comments. Please based on your comments, the necessary corrections have been made. Kindly find attached the responses to your comments. Thanks

Reviewer 2 Report
Manuscript deals with a hot topic and it is interesting and scientifically sound. The IoT/Blockchain pair is an excellent solution for the cyber-security issue. However, due to a number of issues, a major revision is required. In details
consider adding "cyber-security" among keywords
insert paper outline within the Introduction
emphasize in introduction that proposal is located in the IoT context, which is relevant since IoT devices are expected to massively provide data, also very sensitive due to privacy (e.g., healthcare, see <Bennett, Terrell R., et al. "Motionsynthesis toolset (most): A toolset for human motion data synthesis and validation." Proceedings of the 4th ACM MobiHoc workshop on Pervasive wireless healthcare. ACM, 2014.>) or safety (e.g., alarms management within a factory see <Fortino, Giancarlo, Claudio Savaglio, and Mengchu Zhou. "Toward opportunistic services for the industrial Internet of Things." Automation Science and Engineering (CASE), 2017 13th IEEE Conference on. IEEE, 2017.>). To this end, report a clarifying statement and cite the aforementioned papers. Indeed, currently, IoT terms appears in the title (so, readers would expect that is relvant) and only other two times along the paper.
One of the mainstream trend is to move from Cloud to Edge Computing so to provide users with faster IoT services and avoid communication and processing bottlenecks at the Cloud-side (which is far from the data source). Proposal goes in the opposite direction: a comment is therefore required
Discuss about the suitability of the proposed security model's implementation on resource-constrained IoT devices and, in general, about its hw/sw requirements.
Underline the relationship between the proposal and other previous authors' work
At the end of Section 2, provide a paragraph to underline the advancements of the proposal with respect to the surveyed works.
The first 27 lines seem to much didactive and could be removed. By shortening the Introduction, a higher emphasis would be provided on the proposal
perform a deep proof-reading to fix minors (space required at line 237 "model.In our"; "help achieving"->The verb 'help' is used with infinitive: "to achieve" or "achieve"; WSN acronym introduced twice), typos ("presented a an efficient";"et al" -> "et al." at line 95), and re-phrase key statements (e.g., past tense verbs required for the conclusions) for the sake of readability
In conclusion, paper has merit but these issues need to be tackled before proceeding towards a full acceptance
Author Response

(The authors gave the same response as above.)

Reviewer 3 Report
The paper proposes a Proxy Re-Encryption (PRE) scheme with an Inner-Product Encryption (IPE) scheme. Also the authors propose to use a blockchain network with nodes acting as proxy servers performing data re-encryption.
Overall, the paper is interesting and well written. However, for me it is not clear how the proposed scheme would work in a practical implementation.
Especially, the blockchain network has many open issues. What is the format of transactions and blocks? What information is necessary to be stored in transactions and blocks? How does the blockchain network work? (consensus algorithm, number of processing nodes, etc). The authors mentioned smart contracts, but no example is given, neither a brief discussion about this subject. Even if the authors do not address all these issues, some directions have to be given.
Section 5: the authors claim they adopted the schemes presented in [8], so most of the algorithms will be the same. They have to clearly show what is the work presented in [8] and the authors contribution in this section.
There are many recent work on blockchain for e-health systems. Authors could consider discuss some of these related work.
page 10 - line 287 - In our security model (appears twice)
Author Response

(The authors gave the same response as above.)

Round 2
Reviewer 2 Report
Authors have improved the manuscript. Some minors still need to be addressed before submitting the final version:
report the paper outline at the end of Introduction (e.g., In section 2 we..; in section 3 is presented...in section 4, then, we...."
still smooth the style to further improve the paper readability. For example "there are several challenges associated with the IoT. Some of the challenges include providing " -> "there are several challenges associated with the IoT. Some of them provide"; "Setup, KGen, Enrypt and Decrypt are essentially the same as the work presented by [12]" ->"Setup, KGen, Enrypt and Decrypt have been previously presented in [12]"
line 36: instead of "Internet of Things" use "IoT" acronym, already introduced at line 22
replace "Giancarlo Fortino, Wilma Russo, Claudio Savaglio, Mirko Viroli, MengChu Zhou, "Opportunistic 416 cyberphysical services: A novel paradigm for the future Internet of Things", Internet of Things (WF-IoT) 2018 IEEE 4th World Forum on, pp. 488-492, 2018" with its novel, extended journal version "Casadei, Roberto, et al. "Modelling and simulation of Opportunistic IoT Services with Aggregate Computing." Future Generation Computer Systems 91 (2019): 252-262."
Replies provided to reviewer's comments about points 4 and 5 is satisfactory and should be reported in the paper. Here they are reported
"
Point 4: One of the mainstream trend is to move from Cloud to Edge Computing so to
provide users with faster IoT services and avoid communication and processing
bottlenecks at the Cloud-side (which is far from the data source). Proposal goes in the
opposite direction: a comment is therefore required
Response 4: It is true edge computing brings a better satisfaction to IoT devices.
However, the storage of the data is done on the cloud and not much processing is done
by/on the cloud. All processes are executed by the blockchain processing nodes, which
have more computing power than the resource-constrained IoT devices. Moreover, the
protocol still works, be it on the cloud or at the edge since the main focus of this paper
is on the security scheme.
Point 5: Discuss about the suitability of the proposed security model's implementation
on resource-constrained IoT devices and, in general, about its hw/sw requirements.
Response 5: Due to constraints on resources by the IoT devices, the implementation of
the security model, which is part of the computations and processing, is done by the
blockchain network because it has enough processing power. Therefore, there isn’t any
specific hardware/software requirements for the resource-constrained IoT devices.
"
Author Response
Dear Reviewer, please we have made corrections to the manuscript per your suggestions given in the comments. Kindly find attached the responses given to the comments. Thank you.

Reviewer 3 Report
Overall, the paper is interesting and well written. However, for me the authors did not adequately answer the questions of the first review. The main drawback of the paper is the contribution is not clear without more details of the proposal and at least an initial implementation.
In previous review, authors were asked to explain how the blockchain works in their proposal. The authors only answered with some details of the blockchain technology itself and they gave no details of how it would be implemented in the proposal.
All the algorithms presented in section 5 are the same of reference [12]. Considering these observations and the lack of experimental results, I think the contribution of the paper is not enough for a journal paper.
Author Response

(The authors gave the same response as above.)

Round 3
Reviewer 3 Report
For me the contribution is not enough for a journal publication. The idea/proposal is interesting, but it requires more investigation/analysis/implementation to be published.
Author Response
Dear Reviewer, thank you for your userful comments in order to improve this paper. Kindly find attached the response your comment. Thank you, once again.
